# The Concept of "Tradition" in Edmund Husserl

**Rossi Claudio**

World Phenomenology Institute (WPI), Hanover 603, NH 03755, USA; don.claudiorossi@virgilio.it

**Abstract:** That of tradition is a so-called limit problem of Husserlian phenomenology. The text is based on an investigation of the existential and historical character of the formal ego, which implies its temporal and historical stratification and personal constitution. The ego is essentially situated in a spiritual context historically determined by a transmission of ideas and values that have a communitarian character. The fundamental point of this study is to affirm that, in Husserlian thought, tradition, being a transcendental prerequisite of the existential dimension of the formal ego, is consequently a constitutive moment of the human being. The study also brings to light the important concept of "*Vergemeinschaftung*" and provides an interpretation of the theme of the crisis of European mankind, which seems to correspond to an oblivion of its tradition.

**Keywords:** phenomenology; formal ego; historicity; existentialism; intersubjectivity; tradition; community; European Humanity

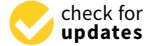



## 1. Explication of the Historical and Existential Sense of the Formal Ego

The radical awareness of "one's own ego" arises from a precise stance, that of overcoming the existential obviousness of the natural attitude and assuming on the contrary a position determined by a fundamental responsibility for the truth. This responsibility represents in Husserl an absolute ethical commitment, and this leads his thinking to the form of a radical formal logic, which he conceives as the science of all sciences, that is, as the science a priori that is transcendently presupposed by any other possible science[1].

One of the fundamental preconditions of this position is that "reality has precedence to every irreality whatsoever, since all irrealities relate back essentially to an actual or possible reality" [6] (§ 64, pp. 168–169). However, this reduction is extremely problematic and requires a difficult process that Husserl calls "transcendental reduction", in which the world in its intrasparency is placed in brackets and the ultimate sources of sense of reality are made apparent. This process is characterized by a radical, even if only initial, de-historicization and dis-individuation of the subjective consciousness in order to reach an ego devoid of any existential concreteness, to an egological form that Husserl calls "pure ego", whose meaning is properly that of being a "centre of functions" or even an "ultimate functional centre" [7] (p. 34). In the manuscripts also occurs the expression "invariant structure" ("*Invariant Struktur*"), which emphasizes the dimension of absolute apodictic-formal invariance of the pure ego[2].

This process of reduction is only functional to the clarification of the logical-formal structures of the intentional processes of meaning constitution which concern every area of the Ego in its existential variety. Therefore, the more in-depth analysis of the formal characteristics of the pure ego leads to the emergence of an essential nexus between formal ego and historically and existentially determined temporality, i.e., leads to the rigorous thematisation of the ego's being as essentially determined as an historical egological being. The topic of existence therefore emerges in the Husserlian thought as an explication of the existential and historical meaning of the formal ego by means of a complete transcendental analytic of the subjective intentional pole. This is a kind of phenomenological radicalism, absolutely necessary in order to understand the essential difference of phenomenology with respect to any form of metaphysics with an idealistic character.

Actually, the transcendental sense of egological ipseity itself entails a permanent and original temporal identity, so that if on the one hand it is a structure that in its formal core is entirely invariant, on the other hand it possesses such invariance in the modality of a temporal identification with itself. It is an absolutely specific flow starting from a transcendental invariance that is temporalized and therefore storicized. As E. Paci points out, it is a distancing of oneself from oneself, a *pro*-jection towards the future, but also at the same time a reversion to the origin. The future, i.e., the historical temporalization of the ego, is always *pro*-jected and not just merely *in*-jected [9] (p. 47).

The fundamental sense of this temporalisation of the ego is, therefore, undoubtedly its transcendence. In the sense of the transcendental ego there is the determination of transcendence: it is never without world. But this means that it "individuates itself", that is, it defines itself by specifying its identity, an identity that historicizes itself in the consciousness of its own diversity, with respect to other identities[3]. Transcendental egoity is not, therefore, a structure, so to speak, "enclosed " in its invariant dimension, but manifests itself as a "subject in the world", among other subjects that are with me in it, for whom is raised the problem of its relationship of real and factual co-existence with the other.

---

1   We want to clarify this important aspect of phenomenological thinking in order to present in a synthetic way the method of analysis followed by Husserl also in relation to questions of a "spiritual" nature such as that of tradition. Husserl elaborated on the necessity of a *doctrine of science* already in the first volume of the *Logische Untersuchungen* on the basis of the recognition that every particular science presupposes "the systematic and theoretical nexus" of the specific knowledge: "The essence of science ( . . . ) involves unity of the foundational connections: not only isolated pieces of knowledge, but their grounded validations themselves, and together with these, the higher interweavings of such validations that we call theories, must achieve systematic unity", [1] (§ 6, p. 18). Therefore, the fact that some cognitive judgments are derived from others through connections regulated by logical laws "not only makes the sciences possible and necessary, but with these also a *theory of science*, a *logic*" (*ivi*) (p. 19). The systematic development of such position occurs in particular in *Phenomenologie und Erkenntnistheorie*, a work of 1917 contained in Hua XXV, where it is clearly declared that the consciousness is *formally*, that is independently of any matter of knowledge, including empirical one, regulated by an infinite variety of *a priori* laws: "An infinite multitude of a priori laws ( . . . ) regulate consciousness before it becomes empirical consciousness" [2] (§ 37, pp. 198–199) (Personal translation). Pure phenomenology emerges thereby as the science that makes these laws explicit and studies them in absolute formality (this is why it is called "pure"). Precisely because the analysis is *pure*, these laws are absolutely *a priori*, i.e., they regulate all possible individual (even empirical) occurrences according to a general essential legality. This set of laws constitutes the authentically "rational" content of science, which forms the indispensable "rational framework" within which all forms of "empirical rationality" can originate and develop. The essence of science is rationality and consequently all empirical rationality is to be considered only of a more precise determination of pure rationality: "The essence of science is rationality. Empirical rationality (scientificity in individual fields) is a more precise definition of pure rationality, which in its pure generality has dimensions of indefinite openness to randomness, which are still subject to empirical rules—within the framework of pure rationality" (*ivi*) (p. 199). (Personal translation). It is clear that at this level of absolute formal generality the pure phenomenology can only be absolutely self-referential, and this means that in process of clarifying the constitution of all the units of consciousness, it also clarifies the constitution of the units that it itself forms: "Thus, phenomenology as science is related to itself as the science of the eidos of pure consciousness. By elucidating the constitution of all units of consciousness, it also elucidates the constitution of the units that it itself naively forms", (*ivi*) (§ 38, pp. 204–205). (Personal translation). This aspect of absolute self-reference appears to be the decisive one to establish phenomenology as a rigorous science and to make it into first philosophy: "The condition of absolute self-reflection ( . . . ) is supposed to provide the absolutely grounded justification of such a rigorous science, and it is to allow phenomenology to come into its own as first philosophy", [3] (p. 257). This science of essence represents therefore a "closed reign of pure rationality" which makes possible, embedded in its frame, the theoretical knowledge of every "particular" reality and of the whole field ("region") to which it belongs. This allows to understand why the method of analysis of each region of being is lately an *eidetic method*, also called "essence research" (*Wesenserforschung*). For Husserl, in fact, judging realities according to the "laws of essence", which are *a priori* laws, is an absolutely necessary task and must be referred to all kinds of realities. Every reality, actually, in Husserl's vision, has its own pure "essence" as its rational content and each of them allows its own exact rational knowledge. And precisely through the "eidetic method" it is brought back to its essential possibility. This is also true in relation to the subject of "tradition", which naturally concerns an aspect related to the spiritual dimension of the human being: "Strict science (*Strenge Wissenschaft*), through the method of evidence of objective truth founded in all its aspects, down to the ultimate roots of knowledge, explores all possible forms and norms of human life and human development, of individual and communitarian life, of development of individual personality and of possible personal communities, specifically of course, in accordance with the general norms, in relation to the formation and development of "genuine" humanity and "genuinely humane" communities" [4] (p. 55). (Personal translation). For a general consideration of the method of the *Wesenserschauung* see Chapter one "*Matter of fact and Essence*" of the Part one "*Essence and eidetic Cognition*" of *Ideas I* [5] (§§ 1–17, pp. 5–33). For a consideration of the method of essence applied to the spiritual sphere of community and humanity see instead [4] (pp. 13–20).

2   "My life of consciousness as mine ( . . . ) bears within itself a wakefulness invariant structure (*trägt in sich eine in der Wachheit invariante Struktur*), which gives the talk of "mine", of me as a consciousness-ego its final reason" [8] (p. 363). (Personal translation).

3   "Penetrating into the transcendental sphere starting from me (*Von mir her im transzendentalen vordringend*), opens up with the question of infinity of temporality, the question of the infinity of the transcendental multiplicity of subjects", *Ms. C 11 I*, p. 6. Quoted in: [9] (p. 46, note 2). (Personal translation).

There are basically two passages that lead to the topic of the historical-existential dimension of the formal ego[4]:

(1) Reduction of the pure ego, as an absolutely invariant formal structure, to my own being, so I discover myself not simply as a formal ego but as a personal one, in a mere temporal form. In this way an overcoming is carried out of the initial position, only theoretically possible, of the a-temporal and a-historical egological ipseity. Therefore, I get myself as "me myself" in a filled-up time, me in my individuated corporeity, in my story. The ego that flows temporally is known as a concrete being of myself within myself, with its determined and individuated history, with its existential concreteness.

(2) On that premise, and at the concomitant time, the other is constituted as an apperceptively detected presence. In other words, in the recognition of my existential concreteness emerges correlatively the existential concreteness of the other and at the same time also our co-existence, which is defined in the sense of a historical *communitarization* (*Vergemeinschaftung*)[5] *of* and *between* different subjects, each of which has its own history[6].

The transcendental in-depth study of the existential thematic of the ego connects, therefore, the monadological individuation of the pure ego to the correlative position of intersubjectivity, so that there is an essential relationship between the monadic egoity and the intersubjective and intra-mundane dimension of its historic-existential "filling". Husserl states with precision in this respect: "Viewed absolutely, every *Ego* has its *history*, and it only exists as subject of a history, of its history. And every communicative community of absolute I's, of absolute subjects—in full concretion, to which belongs the constitution of the world—has its "*passive*" and its "*active*" *history* and only *exists* in this history" [12] (p. 633). The historicity of the ego consists, hence, of two fundamental dimensions: it is the history of the individual ego and, at the same time, the history and community of the egos, in relation to which the life of the individual ego can unfold and have meaning.

In the next paragraph we will try to thematically frame the first of these two dimensions, namely the character of the fully concreteness of the formal ego as a single monad, whose historical factuality is determined as a truly existential genesis. The theme of intersubjectivity, on the other hand, will be addressed only in relation to the problem of tradition, that is, in the sense of transcendental historicization.

## 2. Existential Genesis of the Ego

In a passage of the *IV Cartesian Meditation* Husserl affirms: "the ego grasps himself ( . . . ) also as *I*, who lives this and that subjective process, who live through this and that cogito, *as the same I*" [13] (§ 31, p. 66). In the *Erlebnis*[7], in fact, there is a sort of significant polarization, so that, continues Husserl, "there are always distinguished—in spite of the necessary relatedness to one another—the *mental process itself* and the *pure Ego* pertaining to the mental living. And, again: there are always distinguished the *purely subjective moments of the mode of consciousness* and, so to speak, the rest of the *content of the mental process turned away from the Ego*" [5] (§ 80, p. 191). In this polarization is present the synthesis "of the identical Ego, who, *as the active and affected subject of consciousness*, lives in all processes of

---

4　"The double sense of the I is a key element in Husserl's account of the origin of what it is to have a world (*Welthabe*), and with that the constitution of a shared present of the world (*Mit-gegenwart*) with other subjectivities" [10] (p. 63).

5　The term "*Communitarization*" translates the German "*Vergemeinschaftung*", which expresses an important concept in the issue of tradition. The sense of the word is that of "effecting of communion; establishing of (a) community" [11] (p. 123).

6　For a clear indication of the subdivision of these two fundamental senses of the ego see [8] (pp. 19–20 and p. 434). J. Dodd sums up them in this way: "First (1) the I as the polarization (*Polarizierung*) of all intentional activity; second (2) the I as that into which the entire field of lived experiences itself has matured as personal being or "personality" (*Persönlichkeit*). Both senses of the I work together, in that any instance of (1) forms the material basis for the movement of (2). Thus relative to our example, I am not simply the "I" that performs, the spontaneous center of an act, but a concrete "person" for whom this performance is assimilated into the unity of an existence that is in turn assimilated into a world" [10] (p. 62).

7　The translation in english of '*Erlebnis*', one of the key concepts of phenomenology, is quite difficult. Following D. Cairn '*Erlebnis*' is given in English in first instance through expressions such as "mental process or occurrence, (something lived), (mental) life-process, part of mental life, really immanent process or occurrence", while only in a loose sense and rarely is suitable the term 'experience'. For this reason, according to some scholars, the most appropriate translation should be "*living experience*". In any case, the term '*experience*' referred to '*Erlebnis*' must not be confused with that referred to '*Erfahrung*', which is actually the German term most correctly translated as '*experience*'. Cf. [11] (p. 46)

consciousness and is related, through them, to all object-poles" [13] (§ 31, p. 66). Inasmuch as the pure ego, because of the intentionality of consciousness, is in connection with different objects, in relation to whom it lives different acts in itself, it is possible to identify therein a peculiar form of "concreteness" of the ego: "the pure Ego is further related to Objects in very different modes, according to the type of the act accomplished" [14] (§ 22, p. 104). In this sense, though lacking essential qualities, it possesses modes of "relationship" and "behavior", in reference to which one can speak of a description of the "ways of life" of the ego: the ego gives reason "for a multiplicity of important descriptions precisely with respect to the particular ways in which it is an Ego living in the kinds of mental processes or modes of mental processes in question" [5] (§ 80, p. 191). On this respect, Husserl speaks of a substratum of habituality of the ego, corresponding to the acquisition of certain permanent habits. With regard to such habituality, it is therefore possible to speak of an *ego's life*, to be put in direct relationship with the formation of the personal character of the ego. "The centering Ego", writes Husserl, "is not an empty pole of identity ( . . . ). Rather ( . . . ) with every act emanating from him and having a new objective sense, he acquires a new abiding property. For example: if, in an act of judgment, I decide for the first time in favor of a being and a being-thus, the fleeting act passes; but from now on I am abidingly the Ego who is thus and so decided, "I am of this conviction". As long as it is accepted by me, I can "return" to it repeatedly, and repeatedly find it as mine, habitually my own opinion or, correlatively, find myself as the Ego who is convinced, who, as the persisting Ego, is determined by this abiding habitus or state" [13] (§ 32, pp. 66–67).

We can find here the explanation of a sort of constitution of permanent characteristics that distinguish the pure self of one person in a different way from the pure self of another. In the free acts the ego takes certain positions with respect to objective data, which, as their own, remain in the continuity of the ego, determining this ego in one way or another. In practice, in the course of his "life" the ego undergoes a process of formation, through which it comes to acquiring permanent and personal properties. And it is precisely by virtue of this process that it "overcomes", in a certain way, its strictly formal determination as the "pole-centre" of all the *Erlebnisse* or even as the "identical substratum" of those properties, in order to gain a form of personal identity, by virtue of which one can speak of it as a human person. Husserl writes: "Since, by his *own active generating*, the Ego constitutes himself also as *identical substrate of Ego-properties*, he constitutes himself also as a "fixed and abiding" *personal Ego*" (*ivi*) (p. 67). It is as part of this process that the ego comes to acquire, and then preserve, a "permanent style", which Husserl also calls "personal character": "The Ego shows, in such alterations, an abiding style with a unity of identity throughout all of them: a "personal character"" [*ibid*].

What we have said so far about the existential dimension of the ego concerns the ego as active. The habits, as a matter of fact, are acquired by taking positions (decisions, judgments, etc.) which, as belonging to a permanent ego, become permanent habitus themselves. However, the ego is not only active but also passive: "We find, as *the originally and specifically subjective, the Ego in the proper sense*, the Ego of "freedom", the attending, considering, comparing, distinguishing, judging ( . . . ) and willing Ego: the Ego that in any sense is "*active*" and *takes a position*. This, however, is only one side. Opposed to the active Ego stands the *passive*, and the Ego is always passive at the same time whenever it is active, in the sense of being affected as well as being receptive, which of course does not exclude the possibility of its being sheer passivity" [14] (§ 54, pp. 224–225). This widens the ego's field of life. The ego in relation to objects has not only an active attitude, but also a passive and receptive one, since the objects have an influence on him: "The ego that experiences stimulation from things and appearances, is attracted, and simply yields to the attractive force" (*ivi*) (p. 225).

Husserl draws here all the consequences of the principle of intentionality of consciousness in his conception of the pure self. It's true that "in a certain general sense, the Ego directs itself in every case to the Object, but in a more particular sense at times an Ego-ray, launched from the pure Ego, goes out toward the Object and, as it were, counter-rays issue

from the Object and come back to the Ego" (*ivi*) (§ 22, p. 104). The ego, therefore, does not live only in its acts, inasmuch as it is active, since it possesses an awakening life, but "it experiences excitations from the Objects constituted in the "background": without immediately giving in to them, it allows them to intensify, to knock at the door of consciousness; and then it surrenders ( . . . ) turning from the one Object to the other. In doing so, in the change of its acts, it accomplishes particular turns" (*ivi*) (p. 105).

It is precisely in relation to this dialectical relationship between an active position of the ego and a receptive-passive position towards its objects, that we must speak of a living experience of the ego, so much so that, although it is possible to speak abstractly of a distinction of the pure ego with regard to its acts, in concrete terms "the Ego cannot be thought of as something separated from these lived experiences, from its "life" (*ibid.*). In this "living" relationship of the ego with its intentional objects resides, in fact, a fundamental aspect for the intentional constitutive function: "As ego, I have a surrounding world, which is continually "existing for me"; and, in it, objects as "existing for me" [13] (§ 33, p. 68). In this total concreteness of the ego, objects become, in a certain way, part of the ego itself: "My activity of positing and explicating being, sets up a habituality of my Ego, by virtue of which the object, as having its manifold determinations, is mine abidingly. Such abiding acquisitions make up my surrounding world, so far as I am acquainted with it at the time, with its horizons of objects with which I am acquainted—that is: objects yet to be acquired but already anticipated with this formal object-structure" (*ibid*). It is only in this way that "I exist for myself and am continually given to myself ( . . . ) as "*I myself*" (*ibid.*).

In this very concreteness of one's "life stream" the entire personal potential and capabilities can unfold and each ego explicates itself as a "personal I". It is within such a concreteness of one's "life stream" that each I can unfold one's full potential and personal capacities and develop and manifest itself as a "personal I" (*ich als personales Ich*)[8].

The surrounding world (*Umwelt*) thematic evokes therefore that last reference to the transcendental analytic which Husserl calls "monadic intersubjectivity" and at the same time refers to the acknowledgement of this dimension as "being primary in itself" with respect to the constitution of meaning. "The intrinsically first being", writes Husserl, "the being that precedes and bears every worldly Objectivity, is transcendental intersubjectivity: the universe of monads, which effects its communion in various forms" (*ivi*) (§ 64, p. 156).

### 3. Transcendental Historicity of the Ego

We have seen how in the reflective observation of my "I am" emerges "my being" as an existentially and historically determined factuality: I am a monadic subjectivity in an essential and historically determined relationship with the subjectivities of the others and with a historicity that is the interweaving of my and other people's historicity, an interweaving that has determined me and determines me existentially. In a certain sense, in my ego, as a historically determined egoity, not only the subjectivities of others but also their historicity are implicit: "my own historicity embraces in the way of reality and possibility all the relative monadic historicity"[9]. Moreover, the "monadic totality", understood as the constitutive communitarian sedimentation of the sense of the world, is implicitly present as a decisive factor of my existentially determined being. This is what Husserl explicitly indicates in this passage: "I bear the monadic totality, history and the historical constitution of the world as the "transcendental totality of the world" (monadic

---

8    "Das absolut Seiende: Ich in meiner ständigen Gegenwart, im stehenden Strömen, ich als personales Ich, ich als das schon gezeigte ist in meinem Strömen, mit meinen Vermögen der Wiedererinnerung, der gegenwärtigen und vergangenen Einfühlung, der Vermöglichkeit, in die Einfühlungen einzugehen etc. Von da aus _ist_ die ganze habituelle Vermöglichkeit zu entfalten, korrelativ die für mich seienden Wirklichkeiten und Möglichkeiten, andererseits handelnd fortzuleben mit Anderen und im Rahmen der festen Seinsstruktur die Welt konkret fortzubilden" [8] (p. 441). "The absolute being: me in my constant presence, in the constant vital flow, me as a personal I, the I as already present in my vital flow, with my capacities of recollection, of present and past empathy, of the possibility to enter into empathy etc. From then on, the entire habitual potential _is_ to unfold, correlatively the realities and possibilities that are for me, on the other hand, to live on with others and, within the framework of the fixed structure of being, to continue to form the world on a concrete basis" (Personal translation).

9    *Ms. A VII 11*, p. 16. Quoted in: [9] (p. 52).

totality) in me"[10]. In my "I am" is supposed, as an *a priori* of my being, the *a priori* of my pre-given factuality, which, in turn, is pre-given with and in the pre-giving of the world. This means that my historicity—my existential genesis—which has come to be defined in my determined existential temporality is preceded and founded from another historicity, that inherent in the pre-giving of transcendental subjectivity.

What Husserl wants to assert here is absolutely decisive for the subject of tradition: the being of every transcendental subjectivity finds its constant foundation in a transcendental historicity, which is a sort of traditional genesis that is constantly coming into being and at the same time has already become[11].

A genesis to which each ego spontaneously connects as to a unity of universal apperceptive validity, continously carried forward in him by taking over already formed tradition and with incorporation of new mature universal tradition[12]. This is the way each ego acquire himself as a transcendental I and in community with the other implicit in him transcendental I[13]. For this reason, a "transcendental history" corresponds to the history of each individual ego and precedes it *a priori*[14].

This transcendental primacy of the historicity of the communitarian dimension over the individual ego leads Husserl to affirm that "the ego exists only in the sphere of the we" (*das Ich ist nur im Wir*), provided that this "we" is to be understood not in a restricted sense, limited to particular or even cramped contexts, but in a truly transcendental perspective; i.e., in the sense of a "we" that is relativized in the amplitude of a communitarization (*Vergemeinschaftung*) that tends, in the sense of an infinite task, to really connote itself as an "all-personal We" (*allpersönliches Wir*)[15].

By reason of this "already become" historicity that support and precedes a priori each ego, the *de facto* ego (the *ego-person*) is always in the position of being able to transcend the objective factuality of the world (its naturalness) and to exercise over it not only an *active* but also a *total* praxis: that is, one can define and determine oneself not only starting from oneself, but also from the standpoint of the community. Through this dimension, at the same time *active* and *total*, the subjective praxis of life does not assume a merely solipsistic and "unhuman" dimension, but its temporality can be charged with a truly

---

10　*Ms. A VII 11*, p. 17. Quoted in: [9] (p. 52).

11　"Menschliche Natur und Geschichte wird zur transzendentalen Index der Einheit einer transzendentalen Geschichte, in welcher die transzendentale Subjektivität wesensmässig in jeder transzendentalen Genesis ins Unendliche werdend-geworden ist und in diesem Werdend-gewordensein ihr ständiges Sein hat", *Ms. A V 10*, p. 27. Quoted in [9], (p. 56). "Human nature and history becomes the transcendental index of the unity of a transcendental history, in which transcendental subjectivity—by its very nature—in each transcendental genesis is situated in a process of infinitive becoming-having become [werdend-geworden], and in this process of becoming-to have become [Werdend-gewordensein] has its permanent being" (Personal translation).

12　"Nun, das Immer-Wieder brauche ich in allen Stufen, und die Unendlichkeit ist als Idealisierung selbst eine Ausgestaltung des Immer-Wieder. Es ist universal in mir entsprungen und von mir durch die in mir zur Geltung gekommenen Anderen hindurch erstreckte Einheit der apperzeptiven universalen Geltung—in mir beständig fortgestaltet durch Übernahme schon gebildeter Tradition und Zufügung neuer erwachsener universaler Tradition" [8] (p. 441). "Now, I need the always-again (das Immer-Wieder) in all stages, and infinity, as an idealisation, is itself an implementation of the always-again. It has its universal origin in me and by me through the unity of apperceptive universal validity that come into effect through the others, who came into the fore in me, constantly developed in me by taking over already formed tradition and adding new mature universal tradition". (Personal translation).

13　"Diese kläre ich als phänomenologisches Ich auf, ich gewinne mich als transzendentales und in Gemeinschaft der in mir implizierten anderen transzendentalen Ich" (*ibid*.). "I define this as a phenomenological I; I gain myself as transcendental I, in community with the other transcendental I implied in me". (Personal translation).

14　"Der natürlich menschlichen Historie entspricht eine transzendentale Historie. Die Monaden sind in ihrem primordialen immanent zeitlichen Sein aufeinander bezogen in Form der transzendentalen Bekundung und der gemeinsamen Weltkonstitution, sich als in historischem Menschenzusammenhang findend" (*ivi*) (p. 170). "To the natural human history corresponds a transcendental history. The monads are in their primordial immanent temporal being related to each other in the form of the transcendental communication and of the common constitution of the world, to be found in the historical human context". (Personal translation).

15　"Aber das Ich ist nur im Wir, und notwendig wird das „Wir" ins Unendliche relativiert und im Gemeinschaftlichen, in der Weite der Wir-Bildung auftretenden Vergemeinschaftung zu einem allpersönlichen Wir, das echtes Wir sein soll" (*ivi*) (p.19). "But the "I" is only in the "we", and the "we" is necessarily relativized indefinitely, in the communitarian dimension, in the vastness of the Communitarization that occurs in the formation of the "we", in an all-personal "we", which is supposed to be the real "we". (Personal translation).

"humane" content: the humane ego is self-shaping itself by humanizing itself and shaping the world[16].

We understand, then, how the character of the historical-existential communitarization (*Vergemeinschaftung*) is, therefore, a constitutive factor of the being of the ego as a human "I". The "I" is part of a concrete temporal process that has already begun and that precedes it transcendently. It is part of the history of a world already humanized, which not only expresses but is the historical-existential genesis that preceded it[17].

## 4. Ego's Limits

The crucial importance of the historical-existential dimension of the communitarization (*Vergemeinschaftung*) for the existential genesis of the egological being can be clearly observed with reference to the topic of the ego's limits. The explication of transcendental meaning of the limits of birth and death, emerging as a form of progressive "impoverishment" of the ego up to the point of annulment, brings out, in fact, the inconsistency of a transcendental ego without a sorrounding world (*Umwelt*) that is inter-subjectively founded.

The index of the transcendental ego's participation to materiality and the condition of its being *de facto* in the world is unquestionably the corporeity. It is only through the body that the historical-existential character of the egological being could effectively become a worldly possibility. This possibility, however, lies between two inescapable limits, which are thus likewise the limits of the ego: the beginning and ending, the birth and death.

Birth is phenomenologically conceived as a degree of absolute primordiality of egological life. At this stage the ego acts as a fundamentally "passive" system of perception and instinct, conceivable as a kind of regression of the self-constitutive genesis that leads to a kind of "stripping" of the ego, to a subtraction of everything that pertains to it as a "personal self". It is, in a certain way, an impoverishment of the ego, up to its primary limit, that corresponds to its egological birth [9] (p. 57).

This initial limit has a unique feature throughout its genesis: it has no past in the background, hence does not possess any experiential sedimentation upon which the present experiencing can rely. In this sense, says Husserl, the ego in its extreme original limit is a poor ego (*armseliges Ich*), a merely perceptual corporeity, which does not perceive *objects*, but remains on the level of a mere perception devoid of consciousness of perceived things: it does not have a field of things but only a perceptual domain. This merely perceptive sphere of proximity, anyway, has as its core the living body, whose "*poor ego*", despite being poor, already disposes of the faculties that belong to it[18].

This implies that the ego, in the historic-existential regression, finds as its initial limit the living body as already existent. To this extent, Husserl points out, the birth cannot be conceived as a sudden coming to lightning on the part of the ego, as a mere starting point, but likewise as a *being-in the-beginning*. Here it is not so much a question of an awoken "*I*", but rather of an awakening "*I*", which becomes an *I* of a human being, developing into a

---

16   "Humanisierung ist der ständige Prozess des menschlichen Daseins, Selbsthumanisierung, Sein in beständiger Genesis der Selbstgestaltung und Humanisierung der Welt" [15] (p. 391). "Humanisation is the constant process of human existence, self-humanisation, being in constant genesis of self-formation and humanisation of the world". (Personal translation).

17   "Als schon humanisierte drückt sie beständig ihre frühere Genesis aus. Menschliches Dasein, Sein der menschlichen Welt ( . . . ), ist Sein in beständig lebendiger Geschichte und Sein in sedimentierter Geschichte, die als das ihr immer neues historisches Gesicht hat, dem die Genesis anzusehen, dem sie abzufragen ist" (*ibid*). "As already humanised, it expresses constantly its earlier genesis. Human *Dasein*, being of the human world ( . . . ), is being in a constantly living history and being in a sedimented history, to which, as an always new historical face, the genesis is to be looked at, (to which) it has to be questioned". (Personal translation).

18   "Aber diese Nahsphäre hat noch als *Kern* den Leib, in seiner Zurückbezogenheit auf sich selbst, und hat als Ich das armselig bloss im Leibe waltende armselig Ich mit den zugehörigen Vermögen" [8] (p. 155). "But this near sphere still has the body as its core, in its withdrawal to itself, and has as I the poorly just in the body acting poor I with the respective properties". (Personal translation).

"personal life" and constituting itself in a conscious way as an "*ego for himself*"—a "*personal I*"[19].

The other limit inherent to the ego is that of its *possible ending*. In this case, the present is *pro*-jected onto the future and the mode of the "*I*" is that of a designing "*I*", the "*I*" of the present which opens up to the future, so that the future represents for him before all else his possibility. With respect to the future, the "*I*" defines itself accordingly as "*I can*", on the condition though that the body's original practicality enables that future [10] (p. 156). The failure of the body (illness, injury, death) represents, in fact, a weakening (*depotentiation*) of the future and, consequently, a limitation, to the extreme of annulment, of all the possibilities. It constraints what Husserl refers to as the "practical future", namely the sphere of the formations that could be planned and operatively performed[20].

In this sense, Husserl conceives the final limit of life's genesis, its *end*, as a being totally deprived of all worldly possibilities, a kind of dreamless sleep. Nevertheless, it is only the progressive dissolution of the body and consequently the progressive fading away of the experience that is effectively perceptible. But in this experience is obviously predetermined the possibility of that *limit* in which there is a current powerlessness to experience, i.e., a *not-being-able-to dispose-of-the-world*[21].

This progressive loss of the body brought to the limit, has to be thought as a complete disappearance of the conscious life, of that particular life in which the ego is a pole of

---

[19] "Da wesensmässig die Genesis in Gang ist, sowie das Ich wach ist, dass Ich ein Wahrnehmungsfeld habe, (obschon noch nicht ein Dingfeld), so haben wir als Limes das *erwachende* Ich, das zu einem 'Leben' erwachsend und sich für sich selbst weiter so konstituierend, dass es für sich bewusstseinsmässig—zum Menschen-ich wird" (*ibid*.). "Since by essence Genesis is in progress, as soon as the ego is awake, that I have a field of perception (although not yet a field of things), so we have as the Limes the awakening ego, which develops into a 'life' and continues to constitute himself for himself, so that he becomes for himself conscious—an human-I". (Personal translation).This perspective, that is corroborated by recent neuropsychological research on child psychology, which decisively refutes the "hyper-mechanistic" conception of the modern philosophy of the newborn child's consciousness as a sort of *tabula rasa*, is further analysed in the writing "*The Child. The first Entropathy*" of 1935. Cfr. [15] (pp. 604–606). Recently an authoritative commentary on it, in context with other related texts of Husserlian literature, has been published by Dr. Prof. Ales Bello [16]. In this very interesting text Husserl addresses the issue of the first child's development, even of intrauterine life, and concerns those borderline states in which human consciousness has either not yet formed or seems to disappear. It is of great interest therefore for the theme of tradition, as it highlights the importance of the relationship between the child and the family for the constitution of the primordial surrounding world and the first transmission of a background of information relating to it. For Husserl the consciousness of the infant is by no means a *tabula rasa*. Already in the passage from "*pre-I*" (*Vor-Ich*) to the "*original child*" (*Ur-kind*) the *hyle* (material, basic *substratum*), even though it does not enter into a complete conscientious sphere, receives a pre-structuring and is already a pre-affection dense of meaning. This is the initial development of those potentials that the child possesses to constitute his world. And the first potentiality is *temporality*, which the "*primordial child*" already possesses as a "*pre-dated inner dimension*": that indispensable dimension in which the *Erlebnisse* are inserted in a continuous flow, a sort of inner process, in which time is understood in the Augustinian sense as a "*distentio animi*". The first act is an act from which the infant is affected. Affection is contact with things and spontaneous reaction to them. This structuring of the hyletic dimension, as said, begins already in the *pre-I* phase in the mother's womb, where the sensory fields are present in mutual inclusion with the mother's sensorial fields in a reciprocal sensory experience, without distinction and therefore without the empathic instrument that presupposes two separate configured entities; and it goes on with the first close surrounding world, to which the child turns in the first solicitations. It is essential to note that the potentials are activated by parents who are a community of "*living egos in historical temporality*". Remarkable is furthermore the fact that the first act of the child is that he himself configures the mother, the "first mother", who in the perspective of the child forms and fills his space, while she looks at him, speaks to him, moves between him and things in a physiological eye-mouth triangle. The mother's body, initially perceived as *Körper*, object like others, is "immediately" constituted (naturally in unconscious processes) as *Leib*, i.e., as "living body". Therefore, the infant does not only perceive the mother's eyes, but also her loving gaze. This is where the "first entropathy" (*die erste Einfühlung*) emerges. In this distinction between the child's body and the mother's body it is evident not only that the infant's ego is already formed, but above all that the mother's body, especially in relation to her cradling and embracing hands, is for the child its "first world". The mother functions as a transmission of information, configuring a space with the spatiality of her body.

[20] "Zwar eine gewisse Genesis findet immerfort statt, da dem Ich immer neues weltlich Appercipiertes begegnet, aber mit der Abnahme seiner Kraft verengt sich die Umwelt, die Gegenwärtige, sofern die Sphäre der Zugänglichkeiten sich verkleinert (die relativen 'Entfernungen' werden immer grösser), die Sphäre der Werkgebilde, die noch zu entwerfen und auszuführen wären, sich mindert, womit sich die praktische Zukunft verengert. Aber auch die Vergangenheit engt sich durch Abnahme der Erinnerungskraft, als der Kraft der Verfügbarkeit über die Vergangenheit, ein" [8] (p. 156). "Indeed, a certain genesis continues to take place, since the ego has always new worldly apperceptions, but as its power narrows, the environment, the present time, become narrower. Insofar as the sphere of accessibility reduces itself (the relative 'distances' become ever greater), the sphere of practical operability, which still had to be planned and carried out, decreases, thus narrowing the practical future. But also the past narrows by decrease of the power of memory, as the power of availability about the past". (Personal translation).

[21] "Es erfährt zwar die Umbildung des Leibes als alterndes „Erkranken", aber kein Zerfallen. ( . . . ) Das Zerfallensein des ganzen Leibes kann nicht mehr erfahren sein. Nur ein Limes als der des fortschreitenden Zerfallens ist vorgezeichnet, mit dem Ende: Nichts mehr erhfaren können" (*ivi*) (p. 157). "It experiences the transformation of the body as an ageing "disease", but no decay. ( . . . ) The disintegration of the whole body can no longer be experienced. Only a limit to the progressive decay is predetermined, with its end: nothing more can be experienced". (Personal translation). For the transcendental meaning of birth and death in general see also [17] (pp. 1–26 and 66–82). For an explanation of how Husserl formulates the problem of death from a phenomenological perspective see [10] (esp. pp. 64–68).

identity of the functional intentionality respect to a domain of objects. Thereby Husserl wonders whether such an ego that has ceased to be "worldly" and therefore, at this stage, is conceivable as a fully undifferentiated living in itself, is still the ego of the beginning, the "I that awakens" through the distinct sensation's data, the real ego intentionally characterized as a being oriented towards that or that object, to something different from him. In fact, underlines Husserl, the "I" is essentially a polarization of the life that presupposes counterpoles in a constitutive process of object formation[22].

In others too existential situations one can experience the (possible) "absence of the world" as a form of alienation of the very humanity of the ego's being. In particular Husserl examines that particular situation which he calls "*anguish for the existence*". In such analysis also emerges the inherent connection of the ego with the material body as a transcendental prerequisite of the possible praxis. At this level the *ego-possum* can be outlined as a primary response to need and therefore as *care* (*Sorge*) for today's necessities, which, furthermore, are bound to be reoccurring in the future[23]. Since the eventual unfulfilling of these primary needs poses a very direct threat to the original possibility of the body—and, as a consequence, the impossibility for the ego to act—this dimension can lead to the existential situation of life's anguish, which Husserl describes as "anguish of death" with no representation of the death[24]. Consequently, this original practicality of the ego defines the existence (*Existenz*) as a *care* (*Sorge*), aiming at resolving "the *how*" and not "the *what*" of the planned possibility. Existence, at this level, is determined by two poles: the *hope* of living again and the *threat* of not living anymore[25]. It should be noted how both hope and threat are ultimately directed to the "I can", whose final limit is death, as the limit of the negativity of life: total loss of what is possible, and hence indifference of care (and thereby of threat) and concealment of hope. This is the meaning of death as the *possibility of impossibility*, which, in the end, generates the "anguish for existence" (*Angst um die Existenz*), as a consequence of despair of the possibility of the possible. The negativity of life, however, Husserl points out, is found in every existential situation characterized by the impossibility of a really humane praxis, such as (these are Husserl's examples) the

---

22 "Aber wie ist das denkbar? Und ist dann etwa weder das Ich des Anfangs? Aber dann müsste es "erwachendes" sein, affiziert durch abgehobene Empfindungsdaten. Oder noch nicht waches Ich, wie das des Anfangs vor dem Anfangen (wenn das einen Sinn hat), noch ohne Abgehobenheiten, ein völlig, in sich ungeschiedenes Sein und Leben. Aber hier fragen wir, hat das einen Sinn oder gehört nicht zum Wesen des Ich das Auf-etwashinleben, auf dieses und jenes, auf Unterschiedenes, und ist das Ich nicht eine Polarisierung des Lebens, die Gegenpole in einem konstitutiven Prozess der Gegenstandsbildung voraussetzt?" [8] (p. 158). "But how is that conceivable? And is then neither the ego of the beginning? But then it would have to be "awakening", affected by detached sensory data. Or not yet awake ego, like the ego of the beginning before the beginning (if that makes sense), still without detached sensory data, a completely in itself undifferentiated being and life. But here we ask, does this have a sense or does it not belong to the essence of the ego the aiming-at-something [auf-etwashinleben], to this and that, to differences, and is the ego not a polarization of life, which presupposes counterpoles in a constitutive process of object formation?" (Personal translation).

23 It may be noted that the needs and their satisfaction do not have a properly historical or cultural character. They are not intended to constitute any permanent result: "Die bloss täglichen Bedürfnisbefriedigungen haben keine bleibenden Ergebnisse. Die Zueignung, das Essen der Speise ergibt das Aufbrauchen der Speise", *Ms. A V 5*, p. 169. Quoted in [9] (p. 74, note 5). "The mere daily satisfaction of needs has no lasting results. The labour, the eating of the food results in the consumption of the food". (Personal translation).

24 "Der ungestillt bleibende Hunger—die Ohnmacht des Ich im Handeln: nicht nur Unlust, nicht nur unerfülltes Begehren "inzwischen einmal"—im Spiele des erfüllten, und unerfüllten, bald frei sich auswirkend gehemmt, sondern *Angst* des Daseins. Verzweiflung vor dem Nicht-so-sein, Nicht-fort-sein-können. Die Lebens Angst, die "Todesangst"—ohne Vorstellung des Todes. "Ich vergehe" vor Hunger" [17] (p. 108). "The hunger that remains unquenched—the powerlessness of the ego in action: not only unwillingness, not only unfulfilled desire "once in the meantime"—in the game of the fulfilled and unfulfilled, which soon becomes inhibited acting—but fear of existence. Despair of not being able to be, not being able to continue. The fear of life, the "fear of death"—without the idea of death. "I perish" from hunger". (Personal translation).

25 "Alles Leben in der Hoffnung ist Leben in der Existenzsorge und umgekehrt—wenn eben Existenzsorge Sorge um das Wie und nicht um das Dass der Existenz überhaupt geht" (*ivi*) (p. 520). "All life in the hope is life in the care of existence and vice versa—if the care of existence is concerned with the how and not with the fact of existence". (Personal translation).

existence of the beggar and the lifelong prisoner[26]. On this respect, also the playing can become a gimmick for life satisfaction in place of a truly human way of being[27].

## 5. Traditional Character of the Inter-Subjective Nexus

The wakening life is possible only within a sphere of "human" meanings, which obviously imply the persistence of a social genesis, so that being born is always also an inheritance from a common intersubjective inheritance. The existential genesis of the ego thus involves a *tradere* (*tradieren*), that is, a historical-temporal becoming of the ego in the world, conceived in turn not only as a world of objects, but as a "human" world of inter-subjectively constituted meanings. In this occurs a peculiar mutual identification of the human egoes, both present and former, in which the transmission of meanings (*Sinn-übertragung*) is setup as a legacy of meanings (*Sinn-erbschaft*) [9] (p. 64).

In order to understand this position, it is necessary to consider the concept of spirit as meaningful animation. For Husserl the "donation of meaning" is not an isolated act of the pure subject, but has an ultimately intersubjective foundation. Meaning implies a horizon of meaning, a horizon that includes a plurality of subjective situations that are interwoven into it to form a homogeneous web of meaning that ultimately is the "world". The "world", in this sense, is not simply the object of an isolated perception, but derives from the interchange and agreement of the subjects about their perceptions, and, on the other hand, the subjects are truly related by virtue of the common perceptual field, since there are "things" on which to agree[28]. Consequently, if there is in general a "meaning" it cannot be without the broader underlying unity. One cannot provide meaning without the unity of meaning. In this way the reality of the spirit is made explicit as a "universal nexus" which leads from the history of the individual consciousness to the operating unity of consciences and comes to it as from the founded to the founding. This dialectical relationship Husserl calls "mirroring" [13] (§ 44, pp. 92 ff.). In the "mirroring" the presence of the alter ego is presupposed and implicit. Through it is carried out the recognition of the subject himself as an objective image of the other, whereby the other actually "arises" as an alter ego. Each one shows to the other the naturalness of his being, in which resides the ontological analogy that binds them. But each one is an "expression", that is, an offering of a meaningful situation (we could say of significant "state of affairs"). In virtue of the body-related expression, one can communicate, that is, participate in the "signification" of the other's situation. Discourse, exchange and agreement on "things", situations and meanings becomes possible.

In this process of communication, the *spirit* is generated as a meaningful unity, i.e., as a community of subjects who, in their acting together and in the unity of the human consensus, share a common sense of the *world*. Thus from the plurality of subjects and "things" emerges a validity for all, a unitary evaluation, which gives the world and men a "charge" of intersubjective experience, which places them in a horizon that is no longer "natural" but "historical". Nature has become "world", and the world is historical because the temporal event of the communitarization (*Vergemeinschaftung*) of subjects has been integrated and objectified in a concrete system of material things [18] (p. 212).

---

[26] "Wie in unserer Zeit der Arbeitslogkeit" (*ivi*) (p. 521). "Wenn er "zur Strafe" in einer Isolierzelle nichts zu tun kann, oder (zu) einen für ihn Sinn- und Zwecklosen Tun, Wolle zu zupfen u. dgl., gezwungen sein soll, ohne ernstliche Möglichkeit zu entspringen? Ist ein solches Gefangenleben nicht unmenschlich, nicht von vornherein ent-menschend?" (*ivi*) (p. 522). "If he can do nothing "as a punishment" in an isolation cell, or is forced to do something senseless and pointless for him, like plucking wool, and the like, without any serious possibility of emerging? Isn't such a captive life inhuman, not de-humanizing from the outset?". (Personal translation).

[27] "Wenn der Mensch in solcher Lage sich Spiele erfindet, die ihn beschäftigen, und sich willentlich hineinfinden würde, sich durch Spiele und immer wieder neu durch Spiele zu "erhalten"—wäre das ein menschliches Leben? Ist das nicht sozusagen eine Weise Lebensbefriedigung zu erbetteln, statt zu einer wahrhaft menschlichen Daseinsweise zu kommen?" (*ivi*) (p. 523). "If, in such a situation, a person would invent games to occupy himself and would willingly find his way to "keep" himself through games and over and over again through games—would that be a human life? Isn't that a way of begging for life satisfaction, so to speak, instead of coming to a truly human way of being?". (Personal translation).

[28] "Nature is an inter-subjectivity reality and a reality not just form me and my companions of the moment but for us and for everyone who can have dealing with us and can come to a mutual understanding with us about things and about other people. there is always the possibility that new spirits enter into this nexus" [14] (§ 18, p. 88). Cf. [13] (§ 55, pp. 120 ff.).

Each man, in this sense, is historical because he understands himself in relation to a twofold inter-human past: in a temporal and conscious relationship with the present subjects operating in his living environment and in relation to the generations that succeeded one another before his own, whose trace, in a certain sense its still living "body", is the actual civilization of which he feeds. Tradition, in this way, results to be the intersubjective relationship that characterises the historical dimension of the world and humanity in the world. History is tradition: this means that the essential possibility of history lies in the constitution of an intersubjective temporality, in which we find that deeper, vital and intentional unity, which has equivalent characteristics with the inner consciousness of the time, for which Husserl speak of a synthesis that takes place as a sort of fusion (*Verschmelzung*)[29]. In the intersubjective temporality, the connection between people is constituted between *Erlebnisse* intertwined in a single stream of "general" consciousness.

This supra-individual stream represents the preservation and the development over time of a sense, which constitutes the supporting structure of new intentional formations. Every individual ego arises and is formed in the participation to this "common sense" or "unity of meaning", for which Husserl talks about an actual "sense transmission" (*Sinnübertragung*) or "sense inheritance" (*Sinnerbschaft*), a sort of "spiritual" legacy, which in no way means mere repetions, but a real "intentional unification" (*intentionale Einigung*) established from person to person (*von der Person erbt sich Personales fort auf Personen*)[30]. In this way, the individual consciousness dimension of each ego is such no longer comprehensible only as the mere interiority of the individual, but as a common sphere, as a unity of meaning, and ultimately as the "interiority" of the spirit [18] (p. 212).

Such a "spirit" exists concretely as "animating" the dimension of worldly physicality, thus "transforming" mere nature into a *world*. From this powerful fund of the spirit, which sits at the root of any individual human situations, moves a motivating force all the more powerful the older the unity of a culture, namely the tradition, is (*ivi*) (p. 213). Personal motivations have thus their origin from this sphere of motivations which ultimately resides in the tradition and involves a form of identification with it: "Transcendental is the world constitutive product of transcendently waking subjects as persons in waking contact with each other and in a unity of tradition, in which the world itself is constitutive tradition, beyond the "pauses", birth and death of the individual"[31].

## 6. Being-in-the-Tradition (*Stehen-in-der-Tradition*) as Ontic-Existential Identification of the Ego

The identification with the tradition does not take place at once, but implies a certain process and therefore the temporality. In this sense, it is not so much a matter of learning

---

[29] "Eine Einheit der Verschmelzung verbindet in stetiger Vermittlung das momentan Urimpressionale mit den retentionalen und kontinuierlich verschiedenstufigen Abwandlungen der früheren Impressionen. Diese momentane Simultanverschmelzung ist aber ständig strömend und ist im Strömen beständig sich durch „Zurückschiebung" (Retention) und durch ständig neue Impression inhaltlich wandelnd, aber auch ständig durch Verschmelzung verschmolzene Simultaneinheit in verschmolzene Simultaneinheit überführend, ein Überführen, das selbst ein Zusammenschmelzen ist. Es vollzieht sich natürlich in der Simultaneität höherer Stufe, welche jede Simultaneität mit der soeben gewesenen verbindet, nur dass hier die jeweils momentane Simultaneität die soeben gewesene selbst in sich trägt" [8] (p. 83). "A unity of fusion connects in constant mediation the current primordial impression with retentional and continuously differently graduated variations of earlier impressions. This temporary simultaneous fusion is, however, in a constant state of flux and in flow continually changing in content through "push back" (retention) and constantly new impressions, but also through a constantly transferring of fused simultaneous unit into a new fused simultaneous one, a transfer that is itself a fusion. It naturally takes place in the simultaneity of higher level, which connects every simultaneity with the one that has just been, only that here the current simultaneity carries within itself the one that just occured". (Personal translation). For the important determination of the association occuring in the original temporalizing process as being a mere associative fusion (*urassoziative Verschmelzung*) see [19] (pp. 271–281).

[30] "Assoziation oder Deckung (von Ich miteinander) ist aber Sinnübertragung, Sinnerbschaft, nur dadurch, dass sie Übertragung, Erbschaft von Akthabitualitäten ist, also schließlich vom Ichsein, von der Person erbt sich Personales fort auf Personen. Aber Erbschaft ist nicht Wiederholung, sondern intentionale Einigung, Wandlung, Verdeckung und eben Wandlung durch diese Verdeckung" [8] (pp. 436–437). "However, association or covering (of ego with each other) is a transfer of meaning, an inheritance of meaning, only thereby that is a transfer, an inheritance of acthabitualities, hence ultimately a transfer from the ego, from person to persons of personal matters. However, inheritance is not repetition, but rather intentional unification, transformation, concealment and precisely transformation through this concealment". (Personal translation).

[31] "Transzendental ist die Welt konstitutives Produkt transzendental wacher Subjekte als miteinander im Wachkonnex stehender Personen und in einer Einheit der Tradition, in der Welt selbst sich konstituierende Tradition ist, über die „Pausen", Geburt und Tod des Einzelnen hinaus" [18] (p. 438). (Personal translation).

the tradition, but of *being-in-the-tradition* (*Stehen-in-der-Tradition*); i.e., being in a vital intersubjective nexus, in which the ego derives from others and, as we will see in the next paragraph, from his own community, the principles of his own (indispensable) (re)shaping. It is brought to the fore here again the importance of the influence exerted by parents' heritage and possibly also by the parental levels even earlier in life[32].

This process begins with the same generation, in respect of which one can speak of an *Urtradition* (*original tradition*). The ego that "awakens", since it is generated, is not a mere form of monadic egological being (a formal ego) destined to a subsistence flowing in an atemporal dimension. Rather it is, even in the nocturnal level of purely biological life, a monadic ego already structured, in relation to its personal constitution, by a prospect of possibility made up of character's properties and habitualities which have been transmitted, and in relation to which it shapes itself[33]. However, tradition in its proper sense—i.e., as *transmission*—occurs in historical terms only in the communitarization of waking individuals[34].

It is precisely at this level that the first essential aspect of tradition, that of *identification*, comes into play. To explain this important and very delicate point, Husserl takes his cue from the constitution of the awakened life, which is not at all a constant and inexhaustible being awakened, but on the contrary essentially characterized by a plurality of awakened phases (*Wachperioden*[35]. These *Wachperioden* are individuated in the temporal synthesis, in which is periodically "re-established", beyond the sleeping pauses, the individual singularity of my awakened ego, and consequently the unity of my own duration and the identity of the world around me [20] (p. 335). Now, generalising this consideration, Husserl asserts that "in general it can be said that whenever in the living perceptual temporalisation is constituted such a spatial presence and remembrance reproduces a present with an equivalent, unanimous and over-concordant content with the past, a

---

32 "Daseiendes ist weltlich Seiendes. Weltlich seiend ist hinsichtlich der Vererbung die väterliche oder mütterliche Nachwirkung, das In-Ähnlichkeit-sich-Wiederholen als empirisches Faktum, als empirische Kausalität. Während sonst Selbsterhaltung der Materie statthat, entspringen in dieser Kausalität neue organische „Substanzen" (organische Individuen) und, mit ihnen eins, völlig neuartige personale Einheiten, aber doch ähnliche", (*ivi*) (pp. 438–439). "Existence is worldly existence. With regard to inheritance, worldly being is the paternal or maternal after-effect, the repetition in similarity (*In-Ähnlichkeit-sich-Wiederholen*) as an empirical fact, as empirical causality. While otherwise self-preservation of matter takes place, in this causality new organic "substances" (organic individuals) emerge and, together with them, completely original personal units, but still similar". (Personal translation).

33 "Wir stehen in der Tradition: durch Andere werden wir anders, ihr Personales in uns aufnehmend, es in uns notwendig umbildend. Generativ: es vererbt sich nicht die Leerform monadischer Ichlichkeit, ichlicher Struktur, sondern die vererbten Charaktereigenschaften: wie das? Mit der Erweckung der neuen Monaden ist erweckt oder vorerweckt die latende Habitualität; aber die neue Monade hat eine neue Hyle und die elterliche eine eigene Habitualität (als tote), das alles in sedimentierter Übertragung und sich "mischend", verschmelzend" (*ivi*) (pp. 436–437). "We are in the tradition: through others we become different, taking up their personal qualities [*ihr Personales*] in us, and necessarily transforming them in us. Generative: it is not the empty form of monadic egos, of egoic structure that is inherited, but the inherited character traits: how is that?". (Personal translation). It should be noted: (1) Husserl uses the expression "*I inherit myself*": *Ms. A V 5*, p. 3. Quoted in [9] (p. 65, note 1); (2) he refers the term "*tradition*" to the individual personal ego considered in himself: *Ms. A V 11*, p. 10. Quoted in [9] (p. 65, note 1).

34 "Das Absolute und die einzelnen, absoluten Ich—jedes verweltlicht in der Endlichkeit einer Zeitstrecke; das endliche Ich in der Verkettung seiner Generation, die Generationsunendlichkeit. Die Urtradition der Zeugung; die Zeugenden ihr individuelles Sein tradierend ins erzeugte Individuum; Tradition in der Vergemeinschaftung der wachen Individuen; was mir eigen ist, prägt sich Anderen ein. Deckung der Individuen; assoziative Verschmelzungsprodukte in der Einzelnen und Ineinandertragen des Eigenen und Fremden. So in der vortraditionellen Tradition. Vererbung ursprünglich generative Tradition und Vererbung der gewöhnlichen Tradition (historisch)" [8] (p. 437). "The absolute I and the individual, absolute I (*Das Absolute und die einzelnen, absoluten Ich*)—each one secularised in the finitude of a temporal span; the finite I in the concatenation of its generation, the generational infinity. The primordial tradition of procreation; the procreators transmitting their individual being into the generated individual; tradition in the communitarization of awake individuals; what is my own is "impressed" on others (*prägt sich Anderen ein*). Coverage of the individuals; associative fusion products in the individuals and the interweaving of what is one's own and what is alien. Thus in the pre-traditional tradition (*in der vortraditionellen Tradition*). Inheritance of primordial generative tradition and inheritance of ordinary tradition (historical)". (Personal translation).

35 "Das Einzelsubjekt hat seine Schlafperioden. In jeder Wachheit übersteigt es seine Schlafperiode, in seiner Wachheit frühere Wachheit „wieder" wach machend, und das gehört zum Wesen der Wiedererinnerung—wesensmäßig kann ich „mich" nur wiedererinnern, mich an schon waches Ich, schon in Affektivität und Aktivität Seiendes (wiedererinnern)" (*ivi*) (pp. 437–438). "The individual subject has its sleep periods. In each wakefulness it transcends its sleep period, making earlier wakefulness "awake again", and this belongs to the essence of recollection—by essence I can only remember "me", I can remember myself as being an already awake I, already existing in affectivity and activity". (Personal translation).

synthetic identification must take place. But this does not mean neither "identity" nor difference and equality in the usual sense of the word"[36].

This observation serves Husserl to illustrate the constitutive modality of intersubjective life: the life of a community of subjects and, most importantly, the world that is intentionally referred to them. This is Husserl's argumentation: the intersubjective waking life and the intersubjective world are constituted in the singular-individual periodicity of birth and death, so that the individual singularity of the waking ego itself is a period of a more comprehensive temporal synthesis: "In the world, I find birth and death in the context of generations eternally joining together to form the unity of a generation that connects all people who communicate with each other. In doing so, the possibility of truly separate generations, which only later enter into communication and generative synthesis"[37].

In the genesis of the awakening life, the ego recognises itself in the awakening intersubjectivity, which is outlined as an infinite system of awakening egos, which presupposes a system of "sleeping" egos, of "dead" egos[38]. In this "self-recognition" the identification with the traditionally constituted "handed-down waking life" (*tradierte Wachheit*) takes place[39]. In this sense, points out Piana, "birth and death are to be brought back to the problem of the sense of individual participation to the intersubjective nexus and the internal connection between the intersubjective tradition and the personal tradition" [9] (p. 67).

This is what Husserl clearly expresses with the following words: "We therefore understand intersubjective community, connection as transcendental only insofar as it extends as a world of experience, which is itself an intersubjective tradition, emerging in the self-traditionalisation of every transcendental subject"[40]. It is a sort of primordial identification that constitutes the first sense of the individual's participation to an intersubjective and inter-historical unity, which is not interrupted by birth and death, but on the contrary, precisely through these events, which Husserl terms "connecting members" [*Brückenglieder*],

---

36  "Allgemein ist zu sagen, wenn immer solche räumliche Gegenwart konstituiert ist in lebendiger wahrnehmungsmässiger Zeitigung und die Wiedererinnerung eine ebensolche und inhaltlich einstimmige und übereinstimmende Gegenwart als Vergangenheit, reproduziert, muss eine synthetische Deckung eintreten. Aber das besagt zunächst weder "Identität" noch Verschiedenheit und Gleichheit im gewönlichen Wortsinne", (*ivi*) (p. 421). (Personal translation). For a general and in-depth analysis of the issue of birth and death in relation to the passive constitution see [21], in particular Chapter III, pp. 147 ss.

37  "In der Welt finde ich Geburt und Tod im Zusammenhang der Generationen, die ewig sich zusammenschließen zur Einheit einer Generation, die alle miteinander kommunizierenden Menschen verbindet. Dabei die Möglichkeit wirklich getrennter Generationen, die erst nachher in Kommunikation und in generative Synthese treten. Diese Möglichkeit aber selbst aus meiner Wachheit geschöpft", [8] (p. 438). "In the world, I find birth and death in the context of generations that join together eternally to form the unity of a generation that connects all people who communicate with each other. At the same time, the possibility of actually separate generations which only afterwards enter into communication and in generative synthesis. This possibility, however, is itself derived from my wakefulness". (Personal translation).

38  "Die transzendentale Intersubjektivität—hat sie über die transzendentale Synthese der Wachheiten hinaus (als Parallele der in der Welt seienden Menschengemeinschaft) eine Sphäre der Unwachheit? Ist nicht alle transzendentale Subjektivität, die überhaupt ist, transzendentale wach (wobei wir das ganze sein des transzendentalen—als für sich weltlich konstituierten—Ich als eine Wachheit bezeichnen)? Vielmehr—ein unendliches System wacher Ich, das immerfort ein System "schlafender" voraussetzt, toter—und ist das absolute Universum in dieser Hinsicht ein beständiger Wandel, in dem Leben und Tod, wach-akuelle Ich und tote Ich ihre Funktion üben", (*ivi*) (p. 439). On this point cf. also [17] (pp. 69–73). "Transcendental intersubjectivity—does it have a sphere of unawakeness beyond the transcendental synthesis of wakefulnesses (as a parallel of the human community being in the world)? Is not every transcendental subjectivity at all, transcendentally awake (whereby we designate the whole being of the transcendental—as for itself worldly constituted—I as an awakeness?". (Personal translation).

39  "Aber wie ist das zu denken? Zunächst, wenn sich die Welt transzendental konstituiert als Leistung der je schon verweltlicht seienden transzendentalen Ich, so sagt das, es ist je eine simultane Koexistenz von transzendental wachen Ich mit ihrer Tradition, also für sie vergangener transzendentaler Simultaneitäten und ihrer Leistungen als fortgeltende übernommen; das alles ist tradierte Wachheit. Aber tradiert ist auch das Auftreten neuer „wach" werdender, neu in Kommunikation tretender Subjekte, die aus keiner Tradition her als vergangen gewesen im Horizont der vergangenen Simultaneitäten liegen" [8] (p. 438). "But how is this to be thought? First of all, when the world is transcendentally constituted as the accomplishment of the transcendental I, which is already worldised, this means that it is ever a simultaneous coexistence of transcendentally awake I with its tradition, i.e. of transcendental simultaneities of the past and their achievements taken over as still being valid. All of this is traditional wakefulness. But what is also handed down is the appearance of new subjects who become "awake", who newly enter into communication, who—without any tradition—lie in the horizon of the past simultaneities as having been in the past". (Personal translation).

40  "Wir verstehen also intersubjektive Gemeinschaft, Zusammenhang als transzendentale nur soweit, als Erfahrungswelt reicht, die selbst intersubjektive Tradition ist, in der Selbsttraditionalisierung jedes transzendentalen Subjekts erwachsend" (*ivi*) (pp. 438–439). (Personal translation).

seemingly breaking this unity, it became possible[41]. In this way, states Husserl, "transitoriness is at the same time imperishability, just as a transitory personal action and deed in my life as a past has historical permanence and continues to have effect. So intersubjectivity and its communal life bridges death in a certain way"[42]

## 7. Communitarization (*Vergemeinschaftung*) as Being-in-the-Tradition of One's Own Community

The traditional character of the intersubjective nexus constitutes an intersubjective historicity (*intersubjektive Historizität*) that is not mere "a coexistence of individual being or becoming and of individual histories (traditionalities), but rather a unity of connected humanities, of associations, associations of associations, of intertwining of many forms and intertwining mediators, which have in this being-in-relation (*Verbundenheit*) their historicity, have the unity of a human (personally connected) existence, a historical community life, which cannot be divided into separated individual lives"[43]. There is a continuous mediation between the individual and the community that takes the specific and highly spiritual form of motivation. In concrete terms, in the genesis of its temporal historicity (of its tradition) the individual ego has since ever been in the tradition of its community, in which it is incorporated and which it contributes to constitute. He, states Husserl, "becomes and shapes himself in the inner communitarization of that community ("*als werdender in die Vergemeinschaftung hineinwerdend und sich nach ihr, mit ihr in wechselseitiger Motivation Bildender*") and consequently he carries within himself the genesis that has arisen out of the community ("*in sich die aus Gemeinschaft entsprungene Genesis trägt*"), or, which is the same, he intentionally carries within himself his human educators. This obviously also concern the formation through tradition ("*die Bildung durch Tradition*"), that is, the influence of those men of the same humanity who lived in previous times ("*die Wirkung (sogennante Nachwirkung) von Mitmenschen derselben Menschheit, nur* solcher *früherer Zeiten bedeuten*")"[44]. The traditional nexus thus becomes a recognition of an infinitely open generative nexus towards the past and the future, so that the unity of my personal life is transcended and included in a vaster unity of the generations that preceded me and will follow me. An inter-historical traditional nexus which I contribute to form, becoming myself, when my life takes significant forms, "a tradition in the total universality of the life of humanity"

---

[41] "Die Brückenglieder für die letzte Konstitution sind Geburt und Tod (das Generative)" (*ivi*) (p. 427). "The bridging elements for the last constitution are birth and death (the generative)". (Personal translation). "La naissance, la mort e le sommeil sont comme des interruptions, des courts-circuits à l'intérieur de la continuité du transcendantal. Mais ces interruptions réussissent-elles à rompre la continuité du flux? A lire Husserl, et en particulier les manuscrits sur le temps du groupe C, il semble bien que non: le flux dans sa continuité n'est pas mis en question par ces interruptions mondaines, ni la fatigue, ni par le sommeil ( . . . ), ni meme par la mort pensée par Husserl comme "la soer du sommeil" [21] (pp. 161–162).

[42] "Die Vergänglichkeit ist zugleich Unvergänglichkeit, ähnlich wie eine vergängliche persönliche Handlung und Tat in meinem Leben als vergangenem historisch Bestand hat und Fortgeltung. Also die Intersubjektivität und ihr Gemeinschaftsleben überbrückt in gewisser Weise den Tod" [17] (p. 421) (Personal translation).

[43] "Ein Nebeneinander von personalem Sein und werden und von einzelpersonalen Geschichtlichkeiten (Traditionalitäten), sondern Einheit von verbundenen Menschheiten, von Verbänden, Verbänden von Verbänden, von Verpflechtungen vieler Formen und sich fort verpflechtender Mittelbarkeiten, die in dieser Verbundenheit ihre Historizität haben, Einheit eines menschheitlichen (personal verbundenen) Daseins haben, eines historischen Gemeinschaftsleben, das sich nicht in getrennte Einzelleben zerstücken lässt", *Ms. A V 7*, p. 4. Quoted in [9] (p. 68, note 1). "A juxtaposition of individual being and of individual personal histories (traditionalities), but rather a unity of connected humanities, of associations, associations of associations, of interweavings of diverse forms and interweaving mediatednesses that have their historicity in this interconnectedness, a unity of a human (personally connected) existence, of a historical communal life that cannot be divided into separate individual lives". (Personal translation).

[44] "Der Mensch ist nicht nur in der Gemeinschaft sondern als werdender in die Vergemeinschaftung hineinwerdend und sich nach ihr, mit ihr in wechselseitiger Motivation Bildender, also als jeweils Gewordener, in sich die aus Gemeinschaft entsprungene Genesis trägt, oder, was gleich gilt, seine mitmenschliche Bildner intentional in sich trägt. Das betrifft natürlich auch die Bildung durch Tradition, die Wirkung (sogennante Nachwirkung) von Mitmenschen derselben Menschheit, nur solcher früherer Zeiten bedeuten", *Ms. C 11 III*, p. 8. Quoted in [9] (p. 69, note 2). (Personal translation).

("*eine Tradition in der Universalität des Menschenlebens*")[45]. In this way, I can understand the unity of a history, namely the tradition, which is in the full sense of the word my history, and which is the same as the history of my community, of my people, and so on towards ever larger communitarian unites[46].

It is in this context that the specifically human scope of the generativity is delineated. Actually, the specifically human form of generativity is not purely natural, but is at the same time natural and personal. It is transmission not only of potentialities delimited by the naturalness of the species (animal "*homo*"), but also of a horizon of meanings, and therefore of a *communitarization* founded on a cultural and personal horizon in which the ego finds himself as a human being. Such a horizon is a cultural world, related to the "total personal community", that is, to the people[47].

Precisely because his surrounding world (*Umwelt*) is primarily a cultural world—a world of meanings handed down and shared—the genetic development of the human being is therefore historical and his being is personal: he is called upon to become a person in a history that is both his own and that of the global personal community. Writes Husserl in this regard: "The human being as a person is the subject of a cultural world, which is the correlate of the community of persons in which each individual is aware of himself and is aware of the human cultural world he lives in. The animal does not live, aware of itself, in a cultural world. This obviously implies: man is a historical being, he lives in a "humanity" which is shaped in the historical, history-creating development. The humanity is subjectivity in its role as the bearer of the historical world, an expression which does not mean: the historically living, history-constituting life, but rather the sorrounding world correlate (*das umweltliche Korrelat*), defined as human sorrounding world (*humane Umwelt*), that bears within itself the spiritual meaning of humanity, of the total humanity; and bears it as ontic properties of individual realities and of their ontic historicity, and has this (spiritual) meaning as of human action, of human interests, purposes, systems of human aims"[48].

## 8. Personal and Free Disposition of the Ego towards Tradition

The *being-in-the-tradition*, which we have seen to be the ontic-existential dimension of the ego as a world-historical being, is an historical dimension inasmuch as it is also a *personal* one. The ego as *Dasein* is *person* (and is being called to become person more and more) since it is present in the historical-cultural genesis of its total personal community (people) as a historical subject: determined by a tradition and at the same time determining this tradition. In this sense, Husserl specifies, "the ego is also active ( . . . ) in its essence. It confers on his own volition through its I's activity a modified sense to what has already been experienced as a worldly being, with a specific sense of being by virtue of the past ( . . . ), sketching out the future of identical being"[49].

---

[45] "Ist mein Leben ein rechtes Leben, so bin ich selbst ein bleibender Wert in der Welt (nach meinem Tod bin ich doch weiter das historische Ich, eine Tradition in der Universalität des Menschenlebens), und meine selbstgeschaffenen echten Werte gehen ein in die intersubjektive Welt als bleibende" [17] (p. 421). "If my life is a right life, then I am myself a permanent value in the world (after my death I remain nevertheless the historical I, a tradition in the universality of human life), and my self-made genuine contents and qualities penetrate into the intersubjective world as permanent values". (Personal translation).

[46] Cf. *Ms. C 11 I*, p. 25. Quoted in [9] (p. 69, note 3).

[47] Cf. *Ms. C 11 III*, p. 11. Quoted in [9] (p 71, note 6).

[48] "Der Mensch als Person ist Subjekt einer Kulturwelt, die das Korrelat ist der Personenallgemeinschaft, in der sich jede Person weiss und sich weiss in Bezug auf die humane Kulturwelt, in der sie lebt. Das Tier lebt nicht, sich wissend, in einer Kulturwelt. Dazu gehört offenbar: der Mensch ist ein geschichtiliches Wesen, er lebt in einer "Menschheit", die in dem geschichtlichen, Geschichte schaffenden Werden; sie ist Subjektivität als Träger der geschichtlichen Welt, ein Ausdruck der dann nicht besagt: das historisch lebende, Historie konstituierende Leben, sondern das umweltliche Korrelat bezeichnet, als humane Umwelt, vom Menschen, von der totalen Menschheit her geistige Bedeutung in sich tragend, als ontische Eigenschaften der Realitäten und ihrer ontischen Geschichtlichkeit, als aus dem menschlichen Handeln, aus menschlichen Interessen, Zwecken, Zwecksystemen her diese Bedeutung habend", *Ms. C 11 III*, pp. 12–13. Quoted in [9] (p. 71, note 8). (Personal translation).

[49] "Das Ich ist aber noch in einem anderen Sinne, und zwar wesensmässig aktiv; es erteilt dem schon als weltlich seiend Erfahrenen, schon mit einem bestimmten Seinssinn vermöge der Vergangenheit (eventuell und sogar zumeist durch Apperzeption), die Zukunft identischen Seins vorzeichnend, von sich aus durch seine, des Ich, Aktivität einen veränderten Sinn" [8] (p. 395). (Personal translation).

The activity, the active *praxis*, is therefore an essential dimension of the ego as an historical-worldly being, an inalienable character of being as a person. Through the praxis the ego penetrates the already existing being and modifies its sense, producing a new effect. The historical-cultural world—the *tradition*—is not only a mere actual reality, an effectuality produced by the personal-cultural sedimentation of the past. In relation to the existential position of the ego as active praxis, and therefore as "*I can*", the tradition is not only a permanent effectuality, but also essentially a practical field for possible future effectuations: "Subjectivity, objectified in transcendental self-constitution as human subjectivity, is in the world, and is as such not only getting to know but also shaping the already existing world. The world that is already there, as a world apperceived with a specific sense through the past constitutive processes, the world already predetermined in its possible experience, is not a reality fixed in its existence, but a practical field of action"[50].

This leads us to a consideration of the issue of freedom, or more precisely of the circumstantial sphere in which freedom can concretely express itself in its existential scope. The determination of freedom, in fact, can only be historical, i.e., existentially contextualized, and this means that it is always in relation to a historically determined content. From the previous analyses it is clear that this content comes from the historical horizon of the tradition and not from an isolated individual stance. The very being of situated freedom is thus determined by receiving its contents from tradition.

This condition must not be misinterpreted as a situation of alienation. On the contrary, we actually saw how the topic of inter-subjectivity has clearly shown that the existence of the concrete subject is characterized as a *being-in-the-tradition* and *tradition* has been defined ultimately as a dialectic between the intersubjective exchange of meanings and the assumption of the world in a common overall sense. This clearly illustrates how freedom is fundamentally substantiated and supported by this dialectical movement in its unity, which, in turn, is basically founded and supported by tradition, that is substantiated and supported by tradition, from which, precisely by virtue of free stances of the will of which every ego, in principle, is capable, I can also exempt myself. This is how Husserl expresses this important aspect: "I, who I am as a personal I, as an I who takes free stances (positions), as an I who has a stance derived from my own (and intersubjective) tradition, I (can) exempt myself from all tradition"[51].

It is from the historical moment that every free act receives in concrete terms its determined content, and tradition, in the sense we defined it, is the ground on which it is generated. This in fact very important point is illustrated by D'Ippolito as follows: "In the analysis of the freedom of the person as a worldly and historical being, we discovered the interweaving of personal and interpersonal motivations. The inter-subjective world, as a cultural world, showed to conceal within itself a motivational complex, which is then a unity of meaning based on the experience of generations, strengthened therefore by the countless multiplicity of confirmations. The motivating force of culture, of tradition, operates in history as an incentive to the fulfilment of that experience. And free concrete action is that which is substantiated by historical motivations" [15] (pp. 230–231. Personal translation).

Obviously, however, it cannot be ignored that it is precisely in this historical-traditional configuration of concrete freedom that lies the danger of its possible alienation, which, besides being that of a single individual, is always also the alienation of a whole people. The moment of alienation can thus be a key to interpreting the difficulties of an entire historical phase and a sign of the uneasiness of humanity in its historical course, that is, of its crisis, intended as a widespread and sometimes acute awareness of unease. For

---

[50] "Die Subjektivität, in transzendentalen Selbstkonstitution objektiviert als menschliche Subjektivität, ist in der Welt und ist als das nicht nur Welt kennenlernend, sondern die schon seiende Welt gestaltende. Die jeweils schon seinde als von der subjektivkonstituirenden Vergangenheit her mit Seinssinn apperzipierte Welt, erfahrungsmässig vorgegebene Welt ist nicht die verbleibende seiende Wirklichkeit, sondern praktisches Wirkungsfeld" (*ivi*) (p. 396). (Personal translation).

[51] "Ich, der ich bin als persönliches Ich, als Ich von Stellungnahmen (Stellunghaben), als Ich, das Stellung hat aus eigener (und intersubjektiver) Tradition, enthebe mich aller Tradition" (*ivi*) (p. 225). (Personal translation).

this reason, the problem of alienation is connected by Husserl to the broader issue of the *crisis*: the general formal conditions of the crisis are, to some extent, preliminary to the understanding of the threats and risks that freedom as a *historical* freedom may incur.

## 9. The Spiritual Tradition of European Humanity

Husserl relates freedom to the concept of "authentic praxis", which for him means "authentically rational". In this sense, freedom consists concretely in a praxis that tends towards an increasingly complete "humanization" of the world. Alienation is thus the losing of this spiritual purpose of praxis: it is the praxis separated from the very aim that belongs to it as the man's genuine praxis. Let us try to explain this important aspect.

We have seen how freedom is inserted in the development of tradition, consisting in a cultural movement that possesses a clear cognition of the *telos*, like a value emerging from the past and from past generations, a sort of *spiritual* structure in which a specific mankind, more or less large, has recognized its identity. For humanity (*humanitas*) Husserl means universal communities such as a nation or a global humanity embracing several nations, for example "European" or "Western" humanity[52]. The concept of humanity is defined strictly in relation to that of culture, which is the factor that constitutes a group of individuals in a community of life[53]. And the concept of culture is closely bound up with that of a "living" and lasting tradition and of a spiritual inheritance that presupposes and requires its living transmission through an acceptance and re-comprehension by subsequent generations. So writes Husserl: "By culture we do not mean anything other than the set of actions and operations carried out by associated (*vergemeinschaftet*) men in their continuous activities. Operations that subsist and persist spiritually in the unity of the community's consciousness and its tradition always kept alive. By virtue of the bodily incarnation, of the expression that alienates them from the original creator, they can be experienced in their spiritual sense by whoever is able to re-comprehend (*nach-verstehen*) them. Later, they can again and again become points of irradiation of spiritual effects on ever new generations in the framework of historical continuity"[54]. And it is precisely in this way, Husserl continues, "that everything that is included in a term as "culture" finds its peculiar and essential mode of objective existence and acts, on the other hand, as a constant source of communitarization"[55].

The combination of actions, operations, evaluations and values shared by individuals gathered in their ongoing activities are points of irradiation of spiritual effects on inter-subjectively related individuals and on ever new generations within historical community. The community thus becomes a true "personal subjectivity" animated by a "higher form of life", which allows concrete community actions, that are in the authentic sense "personal"

---

[52] "Universalen Gemeinschaften, die wir "Menschheiten"—eine Nation oder eine mehrere Nationen umfassende Gesamtmenschheit—nennen. Hierher gehort z.B. die "europaische" oder "abendlandische" Menschheit" [4] (p. 21). "Universal communities which we call "humanities"—a nation or a whole humanity embracing several nations. This includes, for example, "European" or "Western" humanity". (Personal translation).

[53] "Eine Menschheit reicht so weit wie die Einheit einer Kultur; zuhöchst einer selbständig abgeschlossenen Universalkultur, die viele nationale Sonderkulturen in sich fassen kann. In einer Kultur objektiviert sich eben eine Einheit tätigen Lebens, dessen Gesamtsubjekt die betreffende Menschheit ist", (*ivi*) (p. 21). "A humanity reaches as far as the unity of a culture; at the highest an independently separated universal culture, which can contain many national specific cultures. In a culture, a unity of active life is objectified, whose total subject is the corresponding humanity". (Personal translation).

[54] "Unter Kultur verstehen wir ja nichts anderes als den Inbegriff der Leistungen, die in den fortlaufenden Tatigkeiten vergemeinschafteter Menschen zustande kommen und die in der Einheit des Gemeinschaftsbewußtseins und seiner forterhaltenden Tradition ihr bleibendes geistiges Dasein haben. Auf Grund ihrer physischen Verleiblichung, ihres sie dem ursprünglichen Schöpfer entäußernden Ausdrucks sind sie in ihrem geistigen Sinn für jeden zum Nachverstehen Befähigten erfahrbar. Sie können in der Folgezeit immer wieder zu Ausstrahlungspunkten geistiger Wirkungen werden, auf immer neue Generationen im Rahmen historischer Kontinuität" (*ivi*) (pp. 21–22). (Personal translation).

[55] "Und eben darin hat alles, was der Titel Kultur befaßt, seine wesenseigentümliche Art objektiver Existenz und fungiert andererseits als eine beständige Quelle der Vergemeinschaftung" (*ivi*) (p. 22). (Personal translation).

operations of the community as such, operations which are carried out consequently according to the aspiration and will of the community itself[56].

The other concept closely related to that of humanity is that of "*ethical man*". What is the life form of the ethical man? The ethical form of life, states Husserl, is characterized by a general aspiration to a truly rational form of self-regulation, which allows to justify oneself in front of reason in all its various dimensions[57]. In this sense, ethical life is closely linked to ethical consciousness as consciousness of the responsibility of reason[58].

It is in relation to the form of ethical life thus defined that the concept of humanity is defined according to the property of *authenticity*, considered by Husserl to be absolutely fundamental. Now for Husserl a humanity is *authentic* because *perfectly rational*. Authentic *humanitas*, in fact, represents the "idea of the "true and authentic man" or rational man" (*wahrhaft vernünftiger Mensch*) [4] (p. 33), which corresponds to the "ideal of absolute personal perfection—perfection of an absolute theoretical, axiological and in every sense practical reason; it is the ideal of a person, as the subject of all personal faculties enhanced in the sense of absolute reason"[59]. Husserl speaks in this case of a mankind that is *awakened* to the *humanitas*, when individuals, sharing the superior value of a community of truly good people, commit themselves to its realization, by creating a spiritual movement that involves others to this superior common goal of a *community of reason* [4] (p. 51 and p. 52); [13] (§ 58, pp. 131–132).

Precisely because of this accentuation of the rational character of the ideal community, so that he speaks, as we have seen, of a "community of reason", Husserl attributes great importance to the role of philosophers and philosophy in general. The philosophy plays indeed a decisive role as a condition of possibility in the development of an authentic community of reason. Philosophers are, in fact, to some extent, the functionaries responsible to delineate the *teleological idea* of an authentically human community inasmuch as it is completely rational, and, in this sense, they represent the "spiritual organism" through

---

[56] "Zeitweise fungiert eine Gemeinschaft vielköpfig und doch in einem hoheren Sinne "kopflos": nämlich ohne daß sie sich zur Einheit einer Willenssubjektivitat konzentrierte und analog wie ein Einzelsubjekt handelte. Sie kann aber auch diese höhere Lebensform annehmen und zu einer "Personalität höherer Ordnung" werden und als solche Gemeinschaftsleistungen vollziehen, die nicht bloße Zusammenbildungen von einzelpersonalen Leistungen sind, sondern im wahren Sinne persönliche Leistungen der Gemeinschaft als solcher, in ihrem Streben und Wollen realisierte" (*ivi*) (p. 22). "At times, a community functions as many-headed and yet in a higher sense "without a head": namely, without being focused into the unity of one subjective will and acting similarly to a single subject. But it can also take on this higher form of life and become a "personality of a higher order" and as such perform communitarian achievements, which are not mere aggregations of individual achievements, but in a true sense personal achievements of the community as such, realised in its efforts and wills". (Personal translation).

[57] "Es bestehen hier wesensmäßige Möglichkeiten für eine Motivation, welche in einem allgemeinen Streben nach einem vollkommenen Leben überhaupt ausmünden, nämlich als einem Leben, das in allen seinen Betätigungen voll zu rechtfertigen wäre und eine reine, standhaltende Befriedigung gewährleistete" (*ivi*) (p. 30). "There are here essential possibilities for a motivation which lead to a general striving for a alltogheter perfect life, namely as a life which would be fully justifiable in all its activities and would ensure a pure, enduring satisfaction". (Personal translation). "Daraus entspringt, in einer möglichen und verständlichen Motivation, ein Wunsch und Wille zu einer vernünftigen Selbstregelung. ( . . . ) Nämlich der Wunsch und Wille, das gesamte eigene Leben, hinsichtlich aller seiner personalen Tätigkeiten im Sinne der Vernunft neu zu gestalten: zu einem Leben aus einem vollkommen guten Gewissen oder einem Leben, das sein Subjekt vor sich selbst jederzeit und vollkommen zu rechtfertigen vermöchte. Wieder dasselbe besagt: zu einem Leben, das reine und standhaltende Zufriedenheit mit sich führte", (*ivi*) (p. 32). "Out of this arises, as a possible and understandable motivation, a will and desire for a rational self-regulation. ( . . . ) Namely, the will and desire to reorganise one's entire life in terms of all one's personal activities in the sense of reason: to a life based on a perfectly good conscience or a life that would be able to justify its own subject before oneself at any time and perfectly. In other words: to a life that led to pure and enduring satisfaction with oneself". (Personal translation).

[58] "Die in einzelnen Fällen bewußt werdende Erkenntnis der Möglichkeit einsichtiger Rechtfertigungen sowie die der Möglichkeit, sein Handeln vorzubereiten und so gestalten zu können, daß es sich nicht nur hinterher und wie zufällig rechtfertigt, sondern, als durch einsichtige Vernunfterwägungen begründet, im voraus die Gewähr seines Rechtes mit sich führt, schafft das Verantwortlichkeitsbewußtsein der Vernunft oder das ethische Gewissen" (*ivi*) (p. 32). "The knowledge, which becomes conscious in individual cases, of the possibility of insightful justifications as well as of the possibility of being able to plan one's actions and to shape them in such a way that they are not only subsequently justified as if by chance, but, as founded on insightful rational considerations, bringimg with them in advance the guarantee of one's right, produces the consciousness of the responsibility of reason or of ethical conscience". (Personal translation).

[59] "Das Ideal absoluter personaler Vollkommenheit—absoluter theoretischer, axiotischer und in jedem Sinne praktischer Vernunft; bzw., es ist das Ideal einer Person, als Subjektes aller im Sinne absoluter Vernunft gesteigerten personlichen Vermogen" (*ivi*) (p. 32). (Personal translation).

which the community becomes aware of its authentic determination[60]. The philosophy will fulfill this task all the more thoroughly, therefore, the more it will be able to elevate itself to the form of *logos*, that is to say, the more it takes the form of a philosophy as a rigorous universal science[61]. It is here that the concept of science is strictly connected to that of philosophy, and both are related to that of the *authentically human mankind*, the one shaping itself in the sense of *authentic humanitas*. In fact, states Husserl, philosophy is a "science inasmuch as it has reached the stage of authentic science, of authentic logos" [4] (p. 55). (Personal translation).

We understand, thus, why Husserl sees in Greek philosophy the paradigmatic model of this philosophy, to the point of speaking of it as an "imperishable glory of the Greek nation". With Platonic dialectics, in fact, for the first time and with an admirable clarity, thinking has been elevated to the fulfilled form of a science concerning the essence of science, that is, to the universality of *logos* as the central normative science of science in general[62]. It is precisely in this way that in the Greek world originates that peculiar culture which has as its fundamental *telos* the attainment of a communitarian life—that is, of a *humanitas*— authentically human, insofar as it is based entirely on rigorous rationality. As Husserl writes textually: "As far as "rigorous" knowledge is concerned, it is a culture of knowledge of a new style and, moreover, a culture that was destined to raise humanity to a new level with regard to its entire life and work. It is the Greeks who, as a consequence of the creation of philosophy in its concise (Platonic) sense, implanted in European culture a universal and of a new kind idea-form, thus assuming the general formal character of a rational culture based on scientific rationality or of a philosophical culture"[63].

Consequently, to these premises, in the eyes of Husserl the humanity that has raised itself to the level of authentic *humanitas* is, historically speaking, that of Europe. It is in the socratic-platonic Greece, which he referred to as "pure Greece", and not in that of the remote Homeric origins, let alone in the naive forms of the first Greek philosophy, that not only the origin but also the fundamental nucleus of European culture stands out, delineating itself in its general formal character as a culture based on an authentic and

---

60 "Die Philosophen sind die berufenen Repräsentanten des Geistes der Vernunft, das geistige Organ, in dem die Gemeinschaft ursprünglich und fortdauernd zum Bewußtsein ihrer wahren Bestimmung (ihres wahren Selbst) kommt, und das berufene Organ für die Fortpflanzung dieses Bewußtseins in die Kreise der "Laien". Die Philosophie selbst ist der objektive Niederschlag ihrer Weisheit und damit der Weisheit der Gemeinschaft selbst; in ihr ist die Idee der rechten Gemeinschaft, also die Zweckidee, welche sich die Gemeinschaft selbst durch ihren Philosophenstand gestaltet hat" (*ivi*) (p. 54). "The philosophers are the vocation representatives of the spirit of reason, the spiritual organ in which the community originally and permanently comes to the consciousness of its true purpose (its true self), and the vocation organ for the propagation of this consciousness into the circles of the "laity". Philosophy itself is the objective embodiment of its wisdom and thus of wisdom of the community itself; in it there is the idea of the right community, i.e. the idea of purpose, which the community itself has formed through its philosophical status". (Personal translation).

61 "Betrachten wir nun die höhere Wertform einer echt humanen, in Selbstgestaltung zu echter Humanitäzt lebenden und sich entwickelnden Menschheit. Es ist diejenige, in der die Philosophie als Weltweisheit die Gestalt der Philosophie als universaler und strenger Wissenschaft angenommen hat, in der die Vernunft sich in der Gestalt des "Logos" ausgebildet und objektiviert hat", (*ivi*) (pp. 54–55). "Let us now consider the higher value form of a genuinely humane mankind living and developing in self-shaping towards authentic humanity. It is the one in which philosophy as worldly wisdom has taken on the form of philosophy as a universal and rigorous science, in which reason has developed and objectified itself in the form of the "logos". (Personal translation).

62 "Der ewige Ruhm der griechischen Nation ist es, nicht nur überhaupt eine Philosophie als eine Kulturgestalt eines rein theoretisehen Interesses begründet zu haben, sondern durch ihr Doppelgestirn Sokrates-Platon die einzigartige Schöpfung der Idee logischer Wissenschaft und einer Logik als universaler Wissensehaftslehre, als normierender zentraler Wissensehaft von der Wissenschaft überhaupt, voIlzogen zu haben. Dadurch erhält der Begriff des Logos als autonomer Vernunft und zunächst theoretischer Vernunft, als des Vermögens eines "selbstlosen" Urteilens, das als Urteilen aus reiner Einsicht ausschließlich auf die Stimmen der Sachen "selbst" hört, seine ursprüngliche Konzeption und zugleich seine weltumgestaltende Kraft" (*ivi*) (p. 83). "The eternal glory of the Greek nation is not only to have founded a philosophy as a cultural form of a purely theoretical interest, but also, through its twin stars Socrates-Plato, to have brought forth the original creation of the idea of logical science and of logic as a universal doctrine of knowledge, as a normative central body of knowledge of science in general. Through this, the concept of logos as autonomous reason and first of all theoretical reason, as the faculty of "self-less" judgement, which as judgement out of pure insight listens exclusively to the voices of things "themselves", obtains its original configuration and at the same time its world-shaping power". (Personal translation).

63 "Handelt es sieh hinsichtlich der "strengen" Wissensehaft um eine Erkenntniskultur eines neuen Stils und zudem um eine Kultur, die dazu bestimmt war, die Menschheit überhaupt hinsichtlieh ihres gesamten Lebens und Wirkens auf eine neue Stufe zu erheben. Die Griechen sind es, die in Konsequenz der Schöpfung der Philosophie in ihrem pragnanten (Platonischen) Sinn der europeischen Kultur eine allgemeine neuartige Formidee eingepflanzt haben, wodurch sie den allgemeinen Formcharakter einer rationalen Kultur aus wissenschaftlicher Rationalität oder einer philosophisehen Kultur annahm" (*ivi*) (pp. 83–84). (Personal translation).

complete rationality[64]. So he clearly states: the "culture based on free reason and, most importantly, on free science striving towards the universal, denotes the absolute teleological idea, the absolute and effective entelechy, which defines the idea of European culture as a unit of development and, if the evaluation is correct, defines it rationally"[65].

## 10. The Crisis of Europe as a Loss of Its Historical and Spiritual Tradition

We have seen how the main characteristic of European humanity consists in the infinite enhancement of human intelligence, which historically has also meant man's hold on reality. In it there is originally an entelechy, which, like an ideal pole, gives meaning and presides in a vital way over its historical processes[66]. Science, in the above mentioned sense of a wisdom (*logos*) authentically realized on the level of pure rationality, is for the European mankind the archimedic point that has enabled it to emerge to a rational life. From this point of view, the power of reason as a distinctly human capacity to shape its own being with respect to a project, has been in it fully expressed. A unitary project that had a precise origin in ancient Greece and that has established a *tradition* that gave unity to this project, by characterizing the history as a rational path and by combining over the centuries various experiences and modifying itself although in its permanence[67].

In this sense, the project of European humanity is inherent within its *cultural tradition*, which has motivated a praxis, assigned to the world of life a law of development, which can be read in institutions, monuments, "traditions", "cultures", in everything that constitutes the world of the "common spirit". It is in this tradition, whose foundation is the authentic *humanitas* as an ethical model founded in an absolutely rational sense, in which the human being is conceived as a person—as a spiritual being—that "Europe" was born as a cultural unity, as an ideal community.

So why the crisis, or rather, where does the crisis come from?

It will not be difficult for us to understand how the crisis of Europe originates from the oblivion of its tradition. It is a real refusal to "re-live" the tradition, to renew and cultivate the values it proposes, or even the refusal to re-discover the unique project, which is also the *telos*, the direction of history. In this process of oblivion not only the value of authentic science and the person is lost, but also that of a spiritual community: of a community united around a ideal *telos*. The crisis of Europe is then the loss of the idea and sense of its spiritual tradition. Ours is the history of the disintegration of a cultural and human unity which has been constituted based on the ideal value of perfect rationality

It must be taken into consideration at this level that in Husserl the historical progression does not correspond to the monadological universe of Leibniz. The Leibnizian

---

64    Of course, as Husserl underlines, that is to be understood *cum grano salis*, inasmuch it is a continuous and progressive shaping: "Bei Platon ist die Idee einer aus freier Vernunft zu gestaltenden Kultur in der pragnanten Form einer philosophisch-wissenschaftlichen Kultur voll ausgestaltet und zudem in reicher systematischer Entfaltung durchdacht. In dieser Pragnanz wirkt sie von nun ab nicht nur in den Studierstuben der Philosophen, sondern gewinnt auch eine starke, die allgemeine Kulturentwicklung bewegende Kraft. Damit ist nicht gesagt daß die griechische, geschweige denn die hellenische Kultur wirklich, in allen Phasen und Schichten, diese Vernunftform erfüllte, die ihr im philosophischen Denken ihre Besten als ihre ideale Form und Norm vorgezeichnet hatten. Genug, daß die formale Norm wirkte und, sei es auch in sehr vermittelten und verflachten Gestaltungen, das Kulturleben bestimmte" (*ivi*) (pp. 108–109). "In Plato, the idea of a culture to be shaped by free reason is fully developed in the pragnant form of a philosophical-scientific culture and, moreover, is thoroughly elaborated in extensive systematic development. In this Pragnanz, from then on, it not only has an effect in the philosophers' study rooms, but also acquires a strong power that moves the general development of culture. This is not to say that Greek, let alone Hellenic, culture really fulfilled, in all phases and levels, this form of reason which its best philosophers had outlined for it as its ideal form and norm. It was enough that the formal norm was effective and determined cultural life, even if in a very mediated and diluted form". (Personal translation).

65    "Kultur aus freier Vernunft und zuhöchst aus freier, ins Universale strebender Wissenschaft bezeichnet dann die absolute Zweckidee, die wirkende absolute Entelechie, welche die Idee der europäischen Kultur als einer Entwicklungseinheit definiert und, wenn die Wertung eine richtige ist, rational definiert" (*ivi*) (p. 109). (Personal translation).

66    "I mean that we feel (and in spite of all obscurity this feeling is probably legitimate) that an entelechy is inborn in our European civilization which holds sway through- out all the changing shapes of Europe and accords to them the sense of a development toward an ideal shape of life and being as an eternal pole" [22] (p. 275).

67    "Here the title "Europe" clearly refers to the unity of a spiritual life, activity, creation, with all its ends, interests, cares, and endeavors, with its products of purposeful activity, institutions, organizations. Here individual men act in many societies of different levels: in families, in tribes, in nations, all being internally, spiritually bound together, and, as I said, in the unity of a spiritual shape. In this way a character is given to the persons, associations of persons, and all their cultural accomplishments which binds them all together. "The spiritual shape of Europe" (*ivi*) (pp. 273–274).

universe is characterised, as known, by continuity and gradualness: from the lowest nature immersed in the sleep of an unconscious cosmos, the realm of the "sleeping monads", the being extends itself to the radiant regions of sensitive consciousness and intelligence, until it takes its own way through the spirit by free decision. Although this aspect is not alien to him, Husserl confronts with resolution the tragic contemporary motif of the crisis, which profoundly changes the Leibnizian model. Mankind through history follows an autonomous path, pursuing the project of humanization, but at the same time runs the risk of failure.

The character of the crisis is entirely a spiritual one. Husserl appears convinced that the European crisis is grounded in an erroneous rationalism which has led to the prevalence of *naturalism* as a basic attitude in science. This tendency has led to the naivety of *objectivism* that took its origin from the natural attitude towards the surrounding worldliness, to which only a dualistic conception of the world would apply, so that nature and the spirit should be considered as realities in the same sense[68]. This has led to the tragic consequence of losing the core idea of the *European spiritual tradition*, namely that of a genuine rationality that founds humanity in all its aspects. So, says Husserl, "precisely this lack of a genuine rationality on all sides is the source of man's now unbearable lack of clarity about his own existence and his infinite tasks" [22] (p. 297).

In the eyes of Husserl there has been a process of mythologizing of the science, which has reduced it fundamentally to technique, and which had as a consequence first of all a split, really fatal, between *logos* and praxis, between science and the world of life, and consequently a naturalization of the person and the historical death of spirituality. The person is reduced to nature and humanity begins to treat itself as an object. Husserl's words on this point are very clear: "Spiritual being is fragmentary. To the question concerning the source of all our difficulties we must now reply: this objectivism, or this psychophysical world-view, in spite of its apparent obviousness, is naively one-sided and has constantly failed to be understood as such. The reality of the spirit as a supposed real annex to bodies, its supposed spatiotemporal being within nature, is an absurdity" (*ivi*) (p. 294).

The foundational tradition of Europe, its spiritual soul, lies instead in the teleological idea of a fully rational *humanitas*, and therefore its failure is that of a rational culture. And, Husserl points out, "the reason for the failure of a rational culture ( . . . ) is in its being rendered superficial, in its entanglement in "naturalism" and "objectivism" (*ivi*) (p. 299). In Husserl's vision, in fact, and this is why he repeatedly hurls himself against them, "naturalism" and "objectivism" represent a conception of the world and life in which the person, the spiritual essence of the human self, no longer has a place.

This crisis has only two possible outcomes: the downfall or rebirth of Europe. The downfall would be marked by a tragic "estrangement from its own rational sense of life, its fall into hostility toward the spirit and into barbarity" (*ibid.*). On the contrary, the rebirth of Europe could occur through "through a heroism of reason that overcomes naturalism once and for all" (*ibid.*). Nevertheless, for Europe, the greatest risk is not the crisis or even the possible downfall "Europe's greatest danger is weariness". And he finishes with notes of hope: "If we struggle against this greatest of all dangers as "good Europeans" ( . . . ) then out of the destructive blaze of lack of faith, the smoldering fire of despair over the West's mission for humanity, the ashes of great weariness, will rise up the phoenix of a new life-inwardness and spiritualization as the pledge of a great and distant future for man: for the spirit alone is immortal" (*ibid.*).

---

[68] "The situation can never improve so long as the objectivism arising out of a natural attitude toward the surrounding world is not seen through in its naiveté and so long as the recognition has not emerged that the dualistic view of the world, in which nature and spirit are to count as realities in a similar sense, though one is built on the other causally, is a mistake" (*ivi*) (p. 297).

## 11. The Issue of Alienation and the Ethical Commitment of Its Overcoming

We have discussed the question od alienation in particular in relation to the issue of the European Crisis. It is necessary now to make some important clarifications on this crucial point in relation to the question of tradition in general.

The possibility of alienation for Husserl pertains to the structural dimension of the human being and of the community. In this regard, he speaks of the transition from a non-authentic form of common life to an authentic one, and even states that it is inherent in the essence of a community (common life) to be "authentically human" only through a transition. The meaning of this transition is that of an evolution leading to a transformation of values (*Wertumwandlung***)**, that can also mean an upheavel and a change of them (*Wertumsturz, -umschlag*), which, in the case of the community, is possible only because it is a practical objective of a common will[69]. In practice, man can and must, both individually and communally, decide freely for the good, and in this sense "tradition" must progress towards the ideal dimension of *ethical man* and in the communitarian sense towards that of true *humanitas*. This position is undoubtedly based on a positive conception of the rational capacity of the human being to identify what is true good and on his volitional determination to pursue it. According to Husserl rationality and practical action are in our power and it is not reasonable, he writes textually, "to deny the possibility of constant ethical progress under the guidance of the rational ideal". He goes so far as to say that the ethical man should be guided, in the pursuit of the rational ideal of a community of truly good people that conforms its entire personal life to the rational norms of the good, by the confidence "that every rational being, if only he knew all this (the rationality of the idea of true *humanitas*), should judge and evaluate in a similar way, and contribute in the same way ( . . . ) to the realization"[70].

Husserl, however, does not ignore the other side of the coin. Man, he states, is essentially sinful because he tends towards the naivety of immediacy: "A life lived in its naive immediateness, without any reflection, leads to sin. As a human being, man is afflicted with original sin, which is part of the essence of man"[71]. By "naivety of immediacy" is meant a style of life marked by irreflection, which gives rise to a life dragged passively by

[69] "Inwiefern es zum Wesen einer Gemeinschaft und eines Gemeinschaftslebens gehört, daß sie die Form einer "echt humanen" Gemeinschaft nur haben kann dadurch, daß sie sich, ausgehend von einer niederen Form, der eines nicht wahrhaft humanen Lebens, etwa gar von der Stufe einer "tierischen" Gemeinschaft oder einer Menschengemeinschaft unwertiger Stufe, zu einer "echt menschlichen" emporgehoben hat: daß also eine "humane" nicht von vornherein da sein, sondern nur durch Entwicklung da sein kann, durch ein Werden, das, ob nun stetig oder diskret, ob passiv oder aktiv oder wie immer, eine Wertumwandlung, einen Wertumsturz, -umschlag vollbringe" [4] (pp. 43–44). "Inasmuch as it belongs to the essence of a community and life's community that it can only have the form of a "genuine human" community by developing from a lower form, that of a life which is not truly human, for example, even from the stage of an "animal" community or of a community of unworthy level, to a "real human" one. In other words, that a "human" community cannot exist from the outset, but can only arise through development, through a becoming, which, whether continuous or discrete, whether passive or active or whatever, brings about a transformation of value, an overthrow of value, a change of value". (Personal translation).

[70] "Indem der individuelle ethische Mensch den Wert eines Vernunftlebens und in bezug auf andere den Wert des Moralischen, den gleichen Wert aller Menschen, soweit sie gut sind oder Gutes im einzelnen wollen etc., für andere erkannt hat und somit den überragenden Wert einer Gemeinschaft aus lauter Guten und eines entsprechenden Gemeinschaftslebens, indem er erkannt hat, daß jeder Vernünftige, wenn er das nur entsprechend erkennen würde, ebenso urteilen und werten müßte und zu der Realisierung ebenso gern nach Möglichkeit beitragen" (*ivi*) (pp. 51–52). (Personal translation). There is no doubt that Husserl is animated by an absolutely positive perspective in the strength of rationality to move the will to act in accordance with right reason. In his vision, if thought, in a rationally evident way, comes to determine and fully clarify the essence and the possibility of its end, procuring to what he calls faith in the ideal of an authentically human community, the foundation of its rational justification, such clarity can provide the will with the necessary strength to carry out a true liberating action. He even speaks of thousands of people convinced by such rationality who finally manage "to move mountains": "Der uns erfüllende Glaube ( . . . ) kann doch nur dann nicht in bloßer Phantasie, sondern in Wirklichkeit "Berge versetzen", wenn er sich in nüchterne rational einsichtige Gedanken umsetzt. ( . . . ) Nur solche Verstandesklarheit kann zu freudiger Arbeit aufrufen, kann dem Willen die Entschlossenheit und die durchsetzende Kraft zu befreiender Tat geben, nur ihre Erkenntnis kann zum festen Gemeingut werden, so daß sich schließlich unter tausendfältiger Mitwirkung der durch solche Rationalität Überzeugten die Berge versetzen, d.i. die bloß gefühlsmäßige Erneuerungsbewegung in den Prozeß der Erneuerung selbst wandelt" (*ivi*) (p. 55). "The faith that fills us ( . . . ) can only then "move mountains" not in mere fantasy, but in reality, when it is translated into sober rationally insightful thoughts. ( . . . ) Only such intellectual clarity can prompt joyful work, can give the will the determination and the assertive power for liberating action, only its insight can become a firm common property, so that ultimately, with the thousandfold cooperation of those convinced by such rationality, the mountains will move, i.e. the merely emotional movement of renewal will be transformed into the process of renewal itself". (Personal translation).

[71] "Ein naiv reflexionsloses Dahinleben führt zur Sünde. Der Mensch ist als Mensch mit der Erbsünde behaftet, sie gehört zur Wesensform des Menschen" (*ivi*) (p. 44). (Personal translation).

inclinations and which does not perform the free act of wanting what is good. This is that life that does not tend towards the level of the ethical man, but remains on an inferior form, which Husserl calls "state of animal naivety" (*Stande tierischer Naivität*)[72]. Man, however, as a subject of self-reflection and of taking a position towards himself, that is, as the subject of an ethical conscience, is subject to a norm of absolute value and must decide freely for what he considers good. In this sense, Husserl speaks of a moral duty to make a "universal choice of life", that of submitting to the categorical imperative that requires him to base all his conduct on good. Man is good only if he adopts this choice in his will. This is how, continues Husserl, "a new form of man arises, the unconditionally superior and required one, that of the man who submits himself to the categorical imperative, who demands a certain form of life from himself and who wants it. This is the type of ethical man and the necessary form of the "real man"[73].

It is clear that in this perspective there is not only the possibility, but even the moral obligation to overcome the alienating forms of life, those forms that can be endured only because one is immersed and remains in the phase of the naive man. The ethical man, quite the opposite, "having been awakened himself out of naivety, wants to give to his own life the form of a good life and to himself that of a good man, of a righteous subject of will, accomplishing the real and authentic goods in every circumstance and in the course of his whole life"[74].

This ethical commitment must of course also apply to the community as a whole. The moral perfection of the ethical man necessarily requires the practical will that everyone can benefit from those goods that are genuinely good. And this can only be attainable in a community that is truly good. Therefore, writes Husserl, for an ethical man to want and pursue the ideal of a truly good community is essential: "It is therefore part of my genuinely human life that I must not only wish for myself to be good, but also for the whole community to be a community of good people, and, as far as I can, take it into my practical circle of will and purpose. Being a true man means wanting to be a true man, and implies wanting to be a member of a "true" humanity, or wanting for the community to which one belongs to be a true community, within the limits of practical possibility"[75].

Taking these premises into account, as far as tradition is concerned, it should be said that Husserl admits and indeed demands as an ethical duty a critical attitude towards tradition(s), but absolutely it cannot be equated with its rejection. For example, when he writes that the ancient model of the Greek idea of science is "not to be taken over blindly from the tradition but must grow out of independent inquiry and criticism" [22] (§ 3, p. 8), this does not mean its denial, but precisely a critical reception of it, which itself stems from a spiritual attitude that is also part of the Greek heritage. And reception

---

[72] "Alle derartigen Lebensformen beruhen auf einem Heraustreten des Menschen aus dem Stande tierischer Naivität. Das ist, das Leben vollzieht sich nicht mehr ausschließlich in naiver Hingabe des Ich an die Affektionen, die von der jeweils bewußten Umwelt ausgehen. Das Ich lebt nicht bloß nach ursprünglichen oder erworbenen Trieben, gewohnheitsmäßigen Neigungen u.dgl., sondern reflektiv wendet es sich ( … ) auf sich selbst und sein Tun, es wird zum sich bestimmenden und wahlenden und, wie im Berufsleben, zu dem sein gesamtes Leben einem reflektiven und allgemeinen Willen unterwerfenden Ich" (*ivi*) (p. 30). "All such life forms are based on the stepping out of the human being from the state of animal naivety. That is, life no longer takes place exclusively in the naïve commitment of the ego to the affections that proceed from the respective conscious sorrounding world. The I does not live merely according to primordial or acquired drives, customary inclinations and the like, but reflectively it turns ( … ) to itself and its actions, it becomes the determining and choice-making ego and, as in professional life, the ego that subordinates its entire life to a reflective and general will". (Personal translation).

[73] "Als Mensch ist er Subjekt der Selbstreflexion, und zwar Stellungnahme zu sich selbst, wertender und praktischer, Subjekt eines "Gewissens", und als das steht er unter einer absoluten Wertnorm: Er soll in jedem Fall nach bestem Wissen und Gewissen sich praktisch entscheiden, er soll nicht nach Neigung passiv sich treiben lassen, soll frei wollen und sich dann frei für das Gute entscheiden, für das, was er erkennend (wenn auch vielleicht irrend) für das Gute erkennt. Nur dann kann er ein "guter Mensch" sein" (*ivi*) (p. 44). (Personal translation).

[74] "Sofern er, aus der Naivität erwacht, sein Leben als ein gutes Leben und sich selbst als guten, als das rechte Willenssubjekt, wahre und echte Güter in jedem Fall und im ganzen Leben verwirklichend, gestalten will" (*ivi*) (p. 46). "Insofar as he, awakened from naivety, wants to shape his life as a good life and himself as good, as the righteous subject of the will, realising true and genuine goods in all cases and in all of his life". (Personal translation).

[75] "Es gehört also zu meinem echt menschlichen Leben, daß ich nicht nur mich als Guten, sondem die gesamte Gemeinschaft als eine Gemeinschaft Guter wünschen und, soweit ich kann, in meinen praktischen Willens-, Zweckkreis nehmen muß. Ein wahrer Mensch sein ist ein wahrer Mensch sein wollen und beschließt in sich, Glied einer "wahren" Menschheit sein wollen oder die Gemeinschaft, der man angehört, als eine wahre wollen, in den Grenzen praktischer Möglichkeit" (*ibid*) (Personal translation).

means that something is transmitted. This is what is rightly pointed out by R. Gasché who writes: "The very spirit of the Greek conception of reason demanded a critical attitude toward the model in question as a heritage bequeathed upon Europe as well as a free and independent reactivation of this heritage" [23] (p. 117). Unfortunately, the same author falls into the prejudicial negative attitude against the factor "tradition" as such, confusing "free and independent reactivation of this heritage" with mere rejection of it when writing: "This universal science embodies the ideal of a community freeing itself precisely from all traditions, and traditionalisms, and shaping itself freely according to insights of reason that are recognizable for their universality" (*idem*). This seems not to be the case. Firstly, as we saw, the insights of the reason are not recognizable simply for their universality, but because they are sustainable before the court of reason itself, i.e., in relation to their unquestionable rationality. Universality of reason must not be confused with abstract universalism, which cuts at the roots the differences historically settled in a living tradition (we will return to this aspect later on in the conclusion). Furthermore, if that statement were correct, strictly speaking it would be necessary to give up that very spirit of universal science which originated in ancient Greece and has been passed down through the centuries as a legacy traditionally settled in the European spirit and sedimented in institutions (e.g., universities) which have their own specific historicity. Husserl's teaching on the existential and therefore traditional dimension of the human being is in fact that the spirit never exists in a mere abstract and existentially de-contextualised generality, but precisely as it is "incarnated" in historically determined traditional and institutional contexts. To exercise the critical operation on such contexts it is not enough, therefore, to appeal, according to the modern prejudicial attitude, to a mere universalisation of principle, but it requires a demanding and laborious effort of analysis of these contexts, which takes into account the inescapable fact that any modification, alteration or rejection of them necessarily entails the assumption of others, who in turn must be evaluated as to their suitability with respect to the spiritual values they should embody and pass on.

In any case, the following important text expresses in a very clear way this relationship, both critical and respectful of reason with regard to tradition(s): "Reason and science do not fight against "tradition" on principle ( . . . ). Reason in its sphere demands a rational tradition, one that can be justified at any time, because as science it is only traditionalised insight. Thus science only fights those traditions which have lost their possible justification or which, in the shifting and reshaping of traditions, now represent an entity (*Gebilde*) that cannot be justified at all. Nonetheless, universal science also seeks to embrace all traditions, on the one hand by establishing the laws for all fields of knowledge as general norms and thus also norms for all traditions. On the other hand, by providing, as a universal history, the universal knowledge, which is equally important for every subject of this history, through which the development of humanity, according to all the main forms of its structures and achievements, becomes comprehensible. In this way all traditions and shifting of traditions can be understood as facts and subsequently, with the aid of other relevant sciences, the evaluation of traditions can be carried out and a distinction can be made between those traditions that are legitimate and effective and those that are illegitimate"[76]

---

[76] "Vernunft und Wissenschaft streiten auch nicht prinzipiell gegen "Tradition" ( . . . ). Die Vernunft in ihrer Sphäre fordert eine rationale Tradition, eine jederzeit zu rechtfertigende, weil sie als Wissenschaft nur traditionalisierte Einsicht ist. So bekämpft Wissenschaft auch sonst nur solche Tradition, die ihre mögliche Rechtfertigung verloren hat oder die in der Verschiebung und Umbildung der Traditionen nun ein Gebilde darstellt, das sich überhaupt nicht rechtfertigen lässt. Doch universale Wissenschaft sucht eben auch alle Traditionen zu umspannen, einerseits dadurch, dass sie in Allgemeinheit die Gesetze für alle Erkenntnisgebiete als allgemeine Normen und damit auch Normen für alle Traditionen aufstellt, andererseits dadurch, dass sie als Universalgeschichte die universale, für jedes Subjekt dieser Geschichte gleich bedeutsame Erkenntnis liefert, durch die das Werden der Menschheit nach allen Hauptgestaltungen ihrer Strukturen und Leistungen verständlich wird und somit alle Traditionen und Verschiebungen von Traditionen als Tatsachen verständlich werden und durch die dann in weiterer Folge unter Behelf anderer in Frage kommender Wissenschaften die Auswertung der Traditionen vollzogen und zwischen rechtmäßig wirksamen und unrechtmäßigen geschieden werden kann", [4] (pp. 179–180). (Personal translation).

## 12. Conclusions

The present study has dealt with a theme that in the context of the Husserlian studies could fall within the so-called *Grenzprobleme*. That of the "tradition" is certainly a secondary topic in the Phenomenology, but of quite a big interest in relation to a complete understanding of the complex phenomenological thinking. In fact, through this theme, a key concept of phenomenology such as that of the world-life is undoubtedly much more comprehensible and a thematic of the very last Husserl, that of the crisis of European humanity, becomes more contextualized.

Husserl shows that all communication between human beings is authentically such when it is placed in a personal context, ultimately that established through a community, which, when it is authentically humanizing, finds its foundation in a living connection with past and future generations. Only on this basis it is possible to speak of a true and authentic humanity. In this sense, every process of alienation of the individual it seems to be related to a process of alienation of the community to which the individual belongs, deriving from a process of individualistic isolation and fragmentation of the different historical and cultural ties that make possible a humanization of life.

In this conlusion we make a synthetic summary of this study and a brief consideration of the interesting links of Husserlian thought on community and tradition with Hegel.

The path followed in this study was first of all based on the survey of the existential and historical character of the formal ego. In Husserl the human being is not limited to the formal dimension of the ego, but implies its temporal structuring and historical stratification, which have both an individual and intersubjective and finally a communitarian dimension in a pregnant sense. To this end, we have talked about of a real existential genesis of the ego, characterized by a progressive acquisition of individual properties and characteristics, which form the personal structure of the individual ego. Each ego has thereby its own personal genesis. This genesis, however, although concerning the singular ego, never occurs in a form of individualistic isolation, but is always existentially and historically contextualised. Such contextualization is radical: it also concerns the first moments of the existence of the ego, its very birth, and includes a necessary, albeit in an active-passive dynamic, *identification* with an already humanised world. My own existential concreteness and that of others are in a necessary mutual relationship of coexistence but also of reciprocal integration with each other. An inter-relation and inter-connection of different subjective histories is determined in a form of association that for Husserl takes the higher ethical form he calls *communitarization* (*Vergemeinschaftung*). That of *Vergemeinschaftung* is a superior ethical community with a specific ontic identity, to the extent that it has its own genesis, i.e., its own historical-temporal existentiality, that constitutes that charge of meaningful values, which is necessarily transmitted, even if otherwise acquired, to the members of the community. This is precisely the broader sense of the concept of *tradition*: the "*unity of conscience of a historical community*". In each ego, insofar as it is a historically determined identity, is therefore implicit not only the historicity of other subjectivities, more or less close to it, but also that of the various communities it belongs to. Each ego receives a traditional inheritance from the sorrounding worlds (*Umwelte*) that are progressively closer to it: from the family, the closer communities to which it belongs, its people. Transcendently, therefore, the ego implies a meaningful world in all those specific determinations that affect it directly or even indirectly. He is thus originally inserted in a *tradition* as a significantly qualified domain, which is transmitted to him and which he is called to embrace in an active-passive attitude.

The fundamental point of our examination was consequently the discovery that in the Husserlian thought tradition is a constitutive moment, that is to say, an original one, of the human being. Tradition then surprisingly emerged as a transcendental prerequisite of the existential dimension of the formal ego. In other words, in the transcendental structure of the ego is detectable an original and constitutive reversion to its own tradition. Thus, for Husserl, against Heidegger, in the ego's being does not prevail the *thrownness* (*Geworfenheit*), but a traditionally determined *grounding*.

What is the specificity of tradition in the context of the intersubjective analysis? With tradition the constitutive relevance of the interpersonal dimension of human beings is determined in a more specific sense as an interrelation of bonds of a "spiritual" nature, bonds that involve the generations of the community in their temporal succession. In this sense, for the human being, besides a general intersubjective sphere, it is constitutive not only the community dimension, for which man has an essentially communitarian character, but also, since it represents one of his entirely original dimensions, to belong to a very precise and distinct "spiritual" community. It is true that in Husserl's vision such a community can take on increasingly larger dimensions. However, it would seem to be understood from the texts we analysed that the need of belonging to a community is not based on a generic level of transcendental determination, but constitutes a necessity always determined in a historical and existential sense. This means that the community dimension of the human being is always first and foremost experienced as belonging to a specific tradition. This tradition may have wider and wider horizons, but the fundamental ties are constituted in the first place by the closer sorrounding worlds (*Umwelte*), "down" to the one that involves more directly the body, that constituted in the first place by the processes of generation and early education in the family. This is why Husserl would agree with Hegel in seeing the family as the first ethical community. This of course does not mean that "tradition" is always good. What is crucial to grasp is, nevertheless, that the meaning of the communitarian dimension as constitutive of the human being lies in the fact that no man exists without community and a correspondent tradition.

The affirmation of tradition as an original dimension of the human being is undoubtedly consistent with the Hegelian principle of the primacy of community over individualism. What was a belief deep-rooted in Hegel that the individual only exists in the space of the community, so that man is essentially communitarian and cannot think of himself except in the community dimension, is justified by Husserl through analyses involving the transcendental moment of inter-subjectivity and the deep level of the temporal genesis of the formal ego, which, as we saw, manifests him as necessarily embedded in a traditional inter-generational bond of communities, more or less close to him in historical and existential terms. The ego is necessarily situated in a "totality and ethical unity", which it could be called, even if the term is not Husserl's, a "community of origin". This is basically the general meaning of the concept of *Vergemeinschaftung*: the necessary existential historicization of the ego in a traditional context.

It is therefore quite apparent that Husserl would agree with Hegel that by nature people precede the individual. One lives first of all in a community and the individual can realize himself only in it. In fact, for Hegel the relational element is not an added factor, but constitutes an original element of a rational system of relations, so that each person is fulfilled through the other, not immediately, i.e., simply according to a principle of mere interchange, but in such a way that the action of each individual person is at the same time as being simply mediated by the form of universality[77]. It is in this sense that the state, rationally understood as a fundamental ethical institution and the foundation of the rational-objective structuring of the human spirit, comes to oppose two fundamentally utilitarian and individualistic interpretations of the communitarian bond in general and of the state in particular: the *contractual* conception, according to which in the state of nature there would only be isolated individuals and society would arise from the contractual aggregation of already formed individuals on the basis of the interests of individuals, which are as such the ultimate purpose for which they are united; and the *naturalistic* conception, for which at the basis of human coexistence there would be the natural law, consistent not in the "strength of what is right and ethical", but "in the accidental violence of nature" [24] (§ 258, pp. 275 ss.). Against these two paradigmatic tendencies, Hegel proposes the famous

---

[77] "The concrete person who, as a particular person, as a totality of needs and a mixture of natural necessity and arbitrariness, is his own end, is one principle of civil society. But this particular person stands essentially in relation (*Beziehung*) to other similar particulars, and their relation is such that each asserts itself and gains satisfaction through the others, and thus at the same time through the exclusive mediation of the form of universality" [24] (§ 182, p. 220).

conception of the state according to its rational realization, which is the real historical response to humanity's need for freedom: "The state in and for itself is the ethical whole, the actualization of freedom" (*ivi*) (p. 279).

From this perspective, a double convergence of Husserl with Hegelian thought can be noted: the emphasis on the essential nature for the human being of the inter-relational bond, which also means its fundamental ontological primariness over the mere individual dimension of his being, and the correspondence, even the coincidence, of the community institutions' ethical quality with their rationality.

But there is another point where agreement of Husserl with Hegel comes out significantly: the importance attributed to the concrete historical dimension of community life and of the traditional sedimentation of communitarian organizations. In Husserl we actually find the overcoming of that abstract intellect which, in the analysis of social structures, stiffens and absolutizes the moment of the individual, thus proving incapable of grasping the nature and importance of historical communal totality. That thought of the abstraction against which the Hegelian social and political thought has vehemently hurled itself. Husserl is so against that basic tendency of the modern world of starting from the abstract individuals, thinking the individual as an isolated monad, which can be morally autonomous, loose from communal moments. One could therefore rightly claim that there cannot be any doubt that Husserl would approve of Hegel's criticism of the contractual and naturalistic conceptions of the state.

With regard to the convergence of Husserl and Hegel on the importance of the traditional dimension *of* and *in* the life of peoples, are of interest the thoughts that Hegel addresses to the constitution as a people's "objectively developed and realized rationality". In his thought the constitution should not consist of an ideal model, developed at table and then imposed through an act of force; nor should it consist of a formal model of state organization, which can then be exported to other social and cultural realities. In general, he affirms, the constitution should not be considered as something that is being produced, but rather as the rational concretization of the way in which a people has come to conceive itself and its forms of political life. Wanting to give a people a "rational" constitution subjectively determined *a priori* would neglect the very moment that makes this constitution more of an *ens rationis*[78]. In this sense, every people has the constitution that historically and culturally is suitable and convenient for them, although the principle of reason, as well as in Husserl, retains its primacy as a formal *telos* of it[79].

Husserl's thinking on tradition and community has also important ethical and social implications. His analyses of the moral obligation to pursue the perfectly rational ideal of the ethical man and true humanity go against those social and political forms based on partial and therefore socially alienating conceptions, such as the "System of needs" that Adam Smith's economic-social conception according to Hegel would entail. The end of reason is precisely that of overcoming those forms of economic organisation that determine social relations marked by extreme individualism, which Hegel did not hesitate to define

---

[78] "Since the state, as the spirit of a nation (*Volk*), is both the law which penetretes all relations within it and also the customs and consciousness of the individuals who belong to it, the constitution of a specific nation will in general depend on the nature and development (*Bildung*) of its self-consciousness; it is in this self-consciousness that its subjective freedom and hence also the actuality of the constitution lie. The wish to give a nation a constitution *a priori*, even if its content were more or less rational, is an idea (*Einfall*) which overlooks the very moment by virtue of which a constitution is more than a product of thought. Each nation accordingly has the constitution appropriate and proper to it. ( . . . ) For a constitution is not simply made: it is the work of centuries, the Idea and consciousness of the rational (in so far as that consciousness has developed in a nation). ( . . . ) The constitution of a nation must embody the nation's feeling for its rights and (present) condition, otherwise it will have no meaning or value, even if it is present in an external sense. Admittedly, the need and longing for a better constitution may often be present in individuals (*Einzelnen*), but for the entire mass (of people) to be filled with such an idea (*Vorstellung*) is quite another matter, and this does not occur until later. Socrates' principle of morality or inwardness was a necessary product of his age, but it took time for this to become (part of) the universal self-consciousness" (*ivi*) (§ 260, 312–313).

[79] "The forms of political constitutions are one-sided if they cannot sustain within themselves the principle of free subjectivity and are unable to conform to fully developed reason" (*ivi*) (§ 273, p. 312).

as barbaric[80]. In this respect, Husserl speaks, as we saw, of the moral duty to pursue authentically rational communities of life. These form of communities necessarily involve the overcoming of those inferior forms of community organisation which in his vision correspond to the stage of "animal" community (*Stufe einer "tierischen" Gemeinschaft*) or human community of unworthy stage (*Menschengemeinschaft unwertiger Stufe*), and the effective establishment of "genuinely humane", i.e., personal, community (*"echt humanen" Gemeinschaft*), in which each ego can be treated and acting as a free person[81].

**Funding:** This research received no external funding.

**Conflicts of Interest:** The authors declare no conflict of interest.

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

81　Cf. note 69.