# Peer review of "The Concept of “Tradition” in Edmund Husserl"

_philosophies, doi:10.3390/philosophies6010001_

Round 1

Reviewer 1 Report

This paper is well written and very provocative for phenomenologists. After reading this article, one can identify better in Husserl's some of the main concepts of Heidegger's phenomenology. The article is a serious contribution to the problem of the intersubjectivity present in the sixth meditation of Husserl's Cartesian Meditations. It is also an apology for the European spirit and tradition.

There are some formal corrections that should be made before publishing:

  • keywords should be inserted after the abstract;
  • a list of references should be inserted at the end;
  • Row 127 “ehich” should become “which”
  • there are some double spaces, which should become one: lines 257, 425, 547.

Author Response

Thank you for your analysis. I tried to improve the article by introducing a more articulated subdivision of the paragraphs and inserting a new one in relation to the problem of alienation and its overcoming according to Husserl (a problem that another reviewer had asked to highlight, given its importance in relation to the issue of tradition). As far as the method is concerned, I tried to illustrate it, especially in its relevance in relation to a "spiritual" theme like the one in question, in a note just at the beginning. I introduced further quotations, trying to translate the most important ones into English, and above all I removed the Italian ones and replaced them with the original German text. I also expanded the conclusion, trying to propose a summary of the work and introducing a small parallel of Husserl's positions on the theme of community and tradition in relation to Hegel (this was also suggested to me). Thank you very much. And best regards. 

Reviewer 2 Report

I am really glad to review this interesting article. Unfortunately, I have to confess that I am not an expert in Husserlian phenomenology. Though I have some experience with the broadly phenomenological tradition, my main research fields are philosophy of mind in analytical tradition and science of consciousness. Thus, some of my comments might be largely peripheral and off-topic. I hope they will be helpful.

First point is about the overall structure of the argument in the paper. While all the analyses in the paper are truly illuminating, where are there epistemic or methodological justification? The author presents ample textual evidence all over the paper, but, of course, it is clear that merely pointing out that Husserl said so would not suffice to support the points the author is trying to make. How Husserl justified, supported, or rationalized what he said must be provided, before or after summarizing his essential analyses. That is, the conceptualizing process is as essential as the very concept of tradition in Husserlian phenomenology. Omitting this process would render all valuable analyses in this paper groundless and unstable.

Second, the multiplicity of tradition may cause several problems in the author's reading of Husserl's analyses. Tradition is neither homogeneous nor neatly unified. Rather, there are many different traditions interacting each other. Strictly speaking, Ego is not born from or developed by one, single tradition. Rather, Ego lives on the intersection of multiple traditions. The point is that there may be, and actually are, tensions or conflicts between these various traditions. There appears to be possible conflicts among traditions of different kind, such as gender, race, age, class, and so on. All these categories have their own history and tradition. Another crucial point is that this actual/possible conflicts between heterogeneous traditions can affect Ego itself. Since the traditional and historical horizons where Ego is born and lives on are structured, Ego should reflect such structure. Further, the actual/possible tensions among multiple traditional, cultural contexts can be transferred into Ego. If so, Ego can be stratified by the multiple traditions it lives on, and there can be friction between those levels of Ego. Even though it is true that Ego has a dimension of tradition, the dimension itself must be articulated further into multiple dimensions. How does Husserl accommodate these issues concerning multiple traditions and their effects on Ego?

Further, some problems remain despite the author's explication about what causes alienation and how it can be cured. One of the central points of the paper is that Ego, in one way or another, is essentially, indeed in constitutive manner, tradition-bounded. In Section 8, dealing with the problem of alienation, the author seems to hint this negative sense of of tradition-boundedness. And, according to the author's interpretation, as far as I can tell, by involving in genuinely free activities or "authentic praxis"(p. 14) one can dodge the risk of being alienated. In other words, alienation can be prevented only when one participates in "a cultural movement that possesses a clear cognition of the telos." (p.14) Though apparently plausible, there are several problems in these diagnosis and prescription. 

1) How can Husserl know that there is such telos? Mentioning 'humanitas', 'telos', and 'spirit', Husserl gives a tempting  grand narrative about us and world. My worry is that this grand narrative, however it is plausible, would be merely a just-so story, if it is not based on evidence or inference. Without such evidence or inference, (the author's reading of) Husserl's analysis is non sequitur.   

2) Even if Husserl knows that there is such telos, how does such telos rescue us from alienation? Whether there is the grand purpose of history of all human kind is one  thing, and whether such purpose will automatically save me from alienation is another. Saying there is certain telos or meaning in this world does not suffice to overcome alienation, or, existential crisis. To do so, one needs to feel it as one's own telos or meaning from one's own perspective, from first-person point of view. The alienation does not come from the mere absence of telos or meaning in the world. We are alienated because there is nothing that is like to have such telos or meaning, even there actually and objectively are such things in the world! In short, it is doubtful that merely adding telos into the world or emphasizing "authentic praxis" would resolve the problem of alienation. 

Further, contrast to the tradition-boundedness of Ego, Ego's active, autonomic capacity to change its own tradition appears to be underrated. Indeed, it is undeniable that we are deeply bounded to our traditions, anchored in our history, and embedded in our culture. However, there is another phenomenological intuition that is equally evident. (it is evident for me, at least) Ego, individually or collectively, escapes from its own tradition, makes its own history, and creates its own culture. Where does this creative power come from? It is hard to find any clue to Ego's escapes and breakthroughs, so to speak.  

There might be several ways to accommodate Ego's escapes and breakthroughs. For instance, Marx famously says "We call communism the real movement which abolishes the present state of things. The conditions of this movement result from the premises now in existence" in his . This kind of real movement resides in Ego, so that how "the conditions  of this movement" can animate Ego to change itself must be explained in order to keep the intuition about Ego's escapes and breakthroughs. Or, there might be some kind of 'hermeneutic circle' between Ego and its traditional and historical horizon. This circular movement may help to understand the dialectics between Ego and its tradition. Further, Hegel's Master-Slave dialectic can be a good candidate. Anyway, I am sure that Husserl already gave an explanation of this Ego-tradition dialectics in his text and the author would easily find it. If the paper is strengthened in this direction, valuable points made in the paper, would become more solid.

Some suggestion from comparative perspective: The author's interpretation of Husserl, in its conclusion, resonates with Hegel's view. Although this paper is neither about Hegel nor about comparative philosophy, comparing Husserl's concept of tradition with Hegel's notion of freedom may render the paper more fruitful and provoke more interesting discussion. It was one of Hegel's main points that without communities, without diverse traditions and social relations, freedom of individual is 'abstract' at best.

Author Response

Thank you very for your profound review. It was really helpful and very stimulating. Also based on it I have made some changes to the text.

1) I tried to improve the article by introducing a more articulated subdivision of the paragraphs and inserting a new one in relation to the problem of alienation and its overcoming according to Husserl. In addition, I have introduced further quotations and translated the most important ones into English, and above all I have removed the Italian ones replacing them with the original German text.

2) As far as the method is concerned, I tried to illustrate it, especially in its value in relation to a "spiritual" theme like the one in question, in a note just at the beginning. In phenomenology the methodological analysis is based on the so-called "vision or evidence of essence" which is, in each specific or "regional" field, the ultimate, absolutely rational justification of what is asserted in relation to that field. This means that at questions like the following that you have proposed: "How can Husserl know that there is such telos?" or "Even if Husserl knows that there is such telos, how does such telos rescue us from alienation?", on a phenomenological level, one can only respond in the following way: through an essence analysis, which offers an ultimate rational justification. Husserl is certain about the power of reason to carry out this task, and all his discourse also about the ideal (and therefore teleological) ethical model of the human being and the community has no other source of justification than that. The same applies to the subject of alienation and its overcoming (including any alienating traditions).

3) This could be important also in relation to your observation, as far as I understand it, about the "epistemic or methodological justification" of how Husserl "justifies, supports, or rationalizes what he sais". Phenomenology, in my opinion, is a method of philosophizing that is strictly analytical, and the argumentative process can only be conceptualized and justified in such a rigorous proceeding. This is the reason why the presentation of a phenomenological topic requires a large number of citations. In any case, I have included in the conclusion a summary of the study, that should be of help for a general survey of it and possibly could also in part respond to your remark about the lack of an "overall structure of the argument in the paper".

4) As far as your notation "contrast to the tradition-boundedness of Ego, Ego's active, autonomic capacity to change its own tradition appears to be underrated", I have subdivided paragraph 7, introducing a new one: "Personal and free disposition of the ego towards tradition". This paragraph, together with the indications given also in the paragraph 11 regarding "alienation and its overcoming", should clarify the problem you have posed. I had to put this paragraph at the end because the topic presupposes concepts presented in paragraphs 9 and 10.

5) I gladly took up your suggestion of a comparative perspective with Hegel, introducing in the conclusion a small parallel of Husserl's positions on the theme of community and tradition with Hegelian thought.

6) In conclusion, I would like to reply to two of your observations which are absolutely central in relation to the study.

a) "Tradition is neither homogeneous nor neatly unified. Rather, there are many different traditions interacting each other. Strictly speaking, Ego is not born from or developed by one, single tradition. Rather, Ego lives on the intersection of multiple traditions".

The study seems to me to suggest with sufficient clarity that there is no single tradition. Tradition to the singular Husserl speaks of only as a transcendental dimension, which specifies, in a certain way, the intersubjective character of the human being, as a historically and existentially determined situationality. This is what I tried to highlight in the conclusion by writing: «The fundamental point of our examination was consequently the discovery that in the Husserlian thought tradition is a constitutive moment, that is to say, an original one, of the human being. Tradition then surprisingly emerged as a transcendental prerequisite of the existential dimension of the formal ego. In other words, in the transcendental structure of the ego is detectable an original and constitutive reversion to its own tradition. Thus, for Husserl, against Heidegger, in the ego's being does not prevail the thrownness (Geworfenheit), but a traditionally determined grounding. [...] This tradition may have wider and wider horizons, but the fundamental ties are constituted in the first place by the closer sorrounding worlds (Umwelte), "down" to the one that involves [...] the processes of generation and early education in the family. This is why Husserl would agree with Hegel in seeing the family as the first ethical community. This of course does not mean that "tradition" is always good. What is crucial to grasp is, nevertheless, that the meaning of the communitarian dimension as constitutive of the human being lies in the fact that no man exists without community and a correspondent tradition».

b) "The actual/possible tensions among multiple traditional, cultural contexts can be transferred into Ego. If so, Ego can bestratifiedby the multiple traditions it lives on, and there can be friction between those levels of Ego. Even though it is true that Ego has a dimension of tradition, the dimension itself must be articulated further into multiple dimensions. How does Husserl accommodate these issues concerning multiple traditions and their effects on Ego?".

To this absolutely important relief naturally Husserl does not respond on a psychological level, but through a rational analysis of what man's being presupposes as an ideal of perfection (which, it should be noted, and in the text I highlight it, is within the reach of a good will supported by perfect clarity about the rational status of the ideal). At this level, Husserl speaks about the rational ideal of the "ethical man" (he even speaks of the ideal of "real man" and the moral duty to pursue such an ideal). And this in turn applies to the community as such. Especially paragraphs 9 and 11 should help to clarify Husserl's position on this point.

Thank you again. My best regards. 

Round 2

Reviewer 1 Report

Well thought article, congratulations!

Reviewer 2 Report

The manuscript has been much improved, both quantitatively and qualitatively. No further comment seems to be needed. I think it is worth publishing.